# Equivariant Matrix Function Neural Networks

**Ilyes Batatia** [1]   **Lars L. Schaaf** [1]   **Gábor Csányi** [1]   **Christoph Ortner** [3]   **Felix A. Faber** [1]

[1] University of Cambridge, UK    [3] University of British Columbia, Canada

## Abstract

Graph Neural Networks (GNNs), especially message-passing neural networks (MPNNs), have emerged as powerful architectures for learning on graphs in diverse applications. However, MPNNs face challenges when modeling non-local interactions in graphs such as large conjugated molecules, and social networks due to oversmoothing and oversquashing. Although Spectral GNNs and traditional neural networks such as recurrent neural networks and transformers mitigate these challenges, they often lack generalizability, or fail to capture detailed structural relationships or symmetries in the data. To address these concerns, we introduce Matrix Function Neural Networks (MFNs), a novel architecture that parameterizes **non-local** interactions through analytic matrix equivariant functions. Employing resolvent expansions offers a straightforward implementation and the potential for linear scaling with system size. The MFN architecture achieves state-of-the-art performance in standard graph benchmarks, such as the ZINC and TU datasets, and is able to capture intricate non-local interactions in quantum systems, paving the way to new state-of-the-art force fields.

## 1 Introduction

Graph Neural Networks (GNNs) have proven to be powerful architectures for learning on graphs on a wide range of applications. Various GNN architectures have been proposed, including message passing neural networks (MPNN) (Gilmer et al., 2017; Battaglia et al., 2018; Kipf & Welling, 2017a; Veličković et al., 2018; Wu et al., 2020; Senior et al., 2018; Hu et al., 2019; Batzner et al., 2022) and higher-order equivariant MPNNs (Batatia et al., 2022b).

MPNNs struggle to model **non-local** interactions both due to computational constraints and oversmoothing (Di Giovanni et al., 2023). Spectral Graph Neural Networks attempt to address the limitation of this kind by encoding the global structure of a graph using eigenvectors and eigenvalues of a suitable operator. These approaches predominantly focus on Laplacian matrices, exploiting the graph's inherent spectral features. Many spectral GNNs apply polynomial or rational filters (Bianchi et al., 2021b; Gasteiger et al., 2018; Wang & Zhang, 2022; He et al., 2021; Defferrard et al., 2016b; Zhu et al., 2021; Kreuzer et al., 2021) to eigenvalues of graph structures, reaching state-of-the-art accuracy on pure graphs tasks. However, these methods often exhibit rigid architectures that require extensive feature engineering, potentially limiting their adaptability to various types of graphs. Moreover, they have been restricted to non-geometric graphs, making them unsuited for molecules and materials.

Traditional neural network architectures, such as recurrent neural networks (Elman, 1990; Hochreiter & Schmidhuber, 1997; Cho et al., 2014; Graves, 2013) and transformers (Vaswani et al., 2017) also face challenges when modeling non-local interactions. While transformers can capture some non-local dependencies through their self-attention mechanisms, they come at a significant computational cost due to their quadratic complexity with respect to the input sequence length, even if existing methods can mitigate their cost Jaegle et al. (2022). Furthermore, transformers lack inherent structural relationships or positional information within input data, necessitating the use of additional techniques, such as positional encodings (Vaswani et al., 2017; Shaw et al., 2018).

In chemistry and material science tasks, some models incorporate classical **long-range** interactions through electrostatics (Grisafi & Ceriotti, 2019; Behler, 2021; Unke & Meuwly, 2019), dispersion, or reciprocal space (Gao & Remsing, 2022; Huguenin-Dumittan et al., 2023; Kosmala et al., 2023). However, no existing architecture effectively address **non-local** interactions, where quantum effects can propagate over extensive distances through electronic delocalization, spin coupling, or

other many-body non-local mechanisms. This is particularly problematic in systems such as large conjugated molecules, amorphous materials, or metals.

Consequently, there is a need for new neural network architectures that can efficiently and accurately model complex non-local many-body interactions, while addressing the limitations of current approaches. We propose **Matrix Function Networks** (MFN) as a possible solution to this challenge. Concretely, we make the following contributions.

- We introduce Matrix Function Networks (MFNs), a new graph neural network architecture able to model non-local interactions in a structured, systematic way.
- We introduce the resolvent expansion as a convenient and efficient mechanism to learn a general matrix function. This expansion can in principle be implemented in linear scaling cost with respect to the size of the input.
- We demonstrate the ability of our architecture to learn non-local interactions on a dataset of challenging non-local quantum systems, where standard GNNs architectures, including those incorporating global attention, fail to give even qualitatively accurate extrapolative predictions.
- We show that MFNs achieve state-of-the-art performance on ZINC and TU graph datasets.

## 2 RELATED WORK

**Overlap matrix fingerprints** (Zhu et al., 2016) introduced overlap matrix fingerprints (OMFPs), a vector of spectral features of an atomic environment or more generally a point cloud. Given a point cloud, an overlap operator (identity projected on an atomic orbital basis) is constructed, and its ordered eigenvalues (or other invariants) are taken as the features of that point cloud. Although a theoretical understanding of OMFPs is still lacking, computational experiments have shown excellent properties as a distance measure (Zhu et al., 2016; Parsaeifard et al., 2021).

**Spectral Graph Neural Networks** Spectral GNNs (Wu et al., 2021) are GNNs that use spectral filters operating on the Fourier decomposition of the Laplacian operator of the graph. Spectral GNNs are categorized by the type of filters they apply to the spectrum of the Laplacian: ChebyNet (Defferrard et al., 2016a) approximates the polynomial function of the Laplacian using Chebychev expansions, GPRGNN (Chien et al., 2021) directly fits coefficients of a fixed polynomial, while ARMA (Bianchi et al., 2021a) uses rational filters.

**Equivariant Neural Networks** Equivariant neural networks are the general class of neural networks that respect certain group symmetries (Bronstein et al., 2021). Notably, convolutional neural networks (CNNs) (LeCun et al., 1989) are equivariant to translations, while $G$-convolutions (Cohen & Welling, 2016; Cohen et al., 2018; Kondor & Trivedi, 2018) generalized CNNs to equivariance of compact groups. Lately, equivariant message passing neural networks (Anderson et al., 2019; Satorras et al., 2021; Brandstetter et al., 2022; Batzner et al., 2022; Batatia et al., 2022b;a) have emerged as a powerful architecture for learning on geometric point clouds. Most of these architectures have been shown to lie in a common design space Batatia et al. (2022a; 2023).

**Hamiltonian Learning** A natural application of equivariant neural network architectures is machine learning of (coarse-grained) Hamiltonian operators arising in electronic structure theory. This task of parameterizing the mapping from atomic structures to Hamiltonian operators is currently receiving increasing interest because of the potential extension in accessible observables over purely mechanistic models. The recent works of Nigam et al. (2022) and Zhang et al. (2022) introduce such parameterizations in terms of a modified equivariant Atomic Cluster Expansion (ACE) (Drautz, 2020), a precursor to the architecture we employ in the present work. Alternative approaches include (Hegde & Bowen, 2017; Schütt et al., 2019; Unke et al., 2021a; Gu et al., 2023).

## 3 BACKGROUND

### 3.1 SPECTRAL GRAPH NEURAL NETWORKS

We briefly review spectral graph neural networks and explain their limitations that our work will overcome. Consider a graph $\mathcal{G} = (X, \mathcal{E})$ with node set $X$ and edge set $\mathcal{E}$. A graph defined purely by its topology (connectivity) is called a pure graph . Let $n$ denote the number of nodes in the graph, and let $\mathbf{A} \in \mathbb{R}^{n \times n}$ be its adjacency matrix. Define a vector of ones as $\mathbf{1}_n \in \mathbb{R}^n$. The degree matrix of the graph is $\mathbf{D} = \text{diag}(\mathbf{A}\mathbf{1}_n)$, and the Laplacian matrix is $\mathbf{L} = \mathbf{D} - \mathbf{A}$. The Laplacian is a

symmetric positive semidefinite matrix and admits a spectral decomposition, $\mathbf{L} = \mathbf{U}\boldsymbol{\Lambda}\mathbf{U}^T$, where $\mathbf{U}$ is an orthogonal matrix of eigenvectors and $\boldsymbol{\Lambda}$ is a diagonal matrix of eigenvalues.

A popular approach to learning functions on graphs is to use convolutional neural networks on the graph. Spectral graph convolutional networks (SGCNs) are a class of graph convolutional networks that use the spectral decomposition of the Laplacian matrix to define convolutional filters. Let $s \in \mathbb{R}^n$ be a function of a graph $\mathcal{G}$ and $t$ be a convolutional filter. SGCNs take advantage of the spectral decomposition of the Laplacian matrix of graph $\mathcal{G}$ to compute the convolution of $s$ and $t$:

$$s' = t \star_{\mathcal{G}} s = \mathbf{U}((\mathbf{U}^T t) \odot (\mathbf{U}^T s)) = f_\theta(\mathbf{L})s, \tag{1}$$

where $f_\theta$ is a matrix function of the Laplacian matrix $\mathbf{L}$ of the graph $\mathcal{G}$ and $\odot$ the Hadamard product. Various works have proposed different matrix functions $f_\theta$, such as polynomial functions (Defferrard et al., 2016a), rational functions (Bianchi et al., 2021a), or neural networks (Wu et al., 2021). Two non-isomorphic graphs can share the same spectrum of their Laplacian operators, while they have different spectra for other graph operators (Johnson & Newman, 1980). Therefore, the use of other graph operators as a basis for matrix functions can be beneficial for learning functions on graphs that are strictly more expressive than Laplacian SGCNs. However, this approach has two main **limitations**.

- **Expressiveness:** Performing only convolutions with a **fixed** graph operator, (usually the Laplacian), limits the expressivity of the model. Choosing the most expressive graph operator requires problem-dependent feature engineering.
- **Lie-group symmetries:** The approaches proposed so far have been restricted to **pure** graphs and in particular do not use additional symmetries for graphs embedded in a vector space. For example, graphs embedded in $\mathbb{R}^3$ often lead to $E(3)$-equivariant learning tasks.

To overcome these limitations, we propose MFNs in Section 4, which allow parameterization of both the graph operator $\mathbf{H}$ and the matrix function $f_\theta$, and can be formulated to preserve the equivariance under all known group actions. Since a learnable operator $\mathbf{H}$ prevents the precomputation of diagonalization of $\mathbf{H}$ during training, we also introduce a method that avoids diagonalization and allows (in principle) linear scaling with the number of nodes.

## 3.2 EQUIVARIANT MESSAGE PASSING NEURAL NETWORK

Equivariant Message Passing Neural Networks (MPNNs) Batzner et al. (2022); Batatia et al. (2022b) are graph neural networks that operate on graphs $\mathcal{G} = (X, \mathcal{E})$ embedded in a vector space $V$. The nodes $x_i \in X$ are no longer only a list of indices, but belong to a configuration space $\Omega$ that extends the vector space $V$. For example, in the atomistic point cloud, $x_i = (i, \boldsymbol{r}_i, \theta_i) \in \Omega := \mathbb{N} \times \mathbb{R}^3 \times \mathbb{Z}$ describing the positions and chemical species of each atom through which the graph is embedded into $V := \mathbb{R}^3$. The case of the **pure** graph can be recovered by setting $\Omega = \mathbb{N}$. We are interested in learning graph maps of the form

$$\Phi \colon \mathrm{msets}(\Omega) \to Z \tag{2}$$

where $Z$ is an abstract target space, usually a vector space and $\mathrm{msets}(\Omega)$ the multi-set of states. As the input is a graph, we impose the mapping to be permutation invariant (invariant under relabeling of the nodes). In many applications, the target properties satisfy additional symmetries: When a group $G$ acts on both $\Omega$ (and, therefore, on $\mathcal{G}$) and $Z$, we say that $\Phi$ is $G$-equivariant if,

$$\Phi \circ g = \rho(g)\Phi \qquad \forall g \in G, \tag{3}$$

where $\rho$ is a representation of the group on the vector space $Z$. A typical strategy is then to embed the nodes $x_i \in X$ into a feature space, where a suitable representation of the group is available.

We represent the state of each node $\sigma_i$ in the layer $t$ of the MPNN by a tuple,

$$\sigma_i^{(t)} = (x_i, \boldsymbol{h}_i^{(t)}), \tag{4}$$

where $x_i$ defines the collection of node attributes of the graph as defined previously and $\boldsymbol{h}_i^{(t)}$ are its learnable features. A forward pass of the network consists of multiple *message construction*, *update*, and *readout* steps. During message construction, a message $\boldsymbol{m}_i^{(t)}$ is created for each node

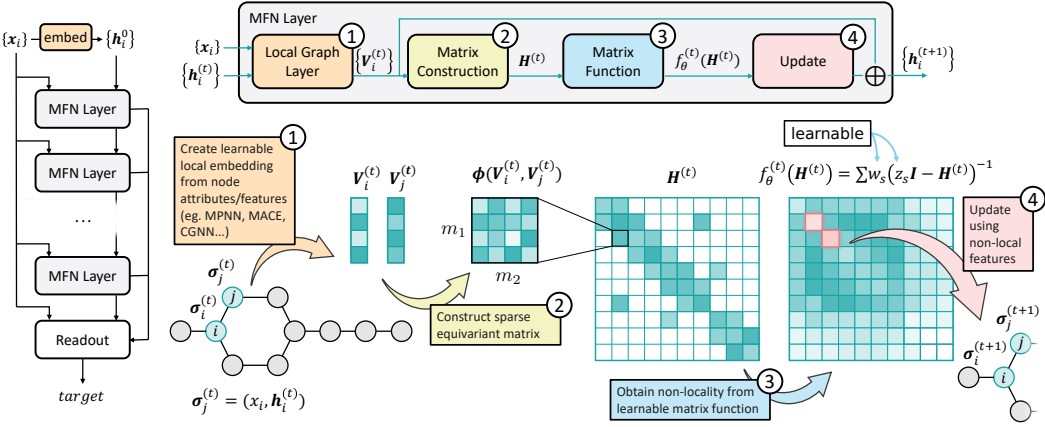

Figure 1: **Matrix function network architecture.** Illustrating matrix construction and non-locality of matrix functions on a molecular graph.

by pooling over its neighbors,

$$\boldsymbol{m}_i^{(t)} = \bigoplus_{j \in \mathcal{N}(i)} M_t\big(\sigma_i^{(t)}, \sigma_j^{(t)}\big), \quad \boldsymbol{h}_i^{(t+1)} = U_t\big(\sigma_i^{(t)}, \boldsymbol{m}_i^{(t)}\big), \quad \Phi(\mathcal{G}) = \phi_{\text{out}}\left(\Big\{\big\{\mathcal{R}_t\big(\sigma_i^{(t)}\big)\big\}_i\Big\}_t\right).$$

(5)

where the individual operations have the following meaning:

- $M_t$ is a learnable message function and $\bigoplus_{j \in \mathcal{N}(i)}$ is a learnable, permutation invariant pooling operation over the neighbors of atom $i$ (e.g., a sum);
- $U_t$ is a learnable update function, transforming the message $\boldsymbol{m}_i^{(t)}$ into new features $\boldsymbol{h}_i^{(t+1)}$;
- $\mathcal{R}_t$ is a learnable node readout, mapping node states $\sigma_i^{(t)}$ to the per-node outputs;
- $\phi_{\text{out}}$ is a global readout map, typically $\phi_{\text{out}}(\{\{\mathcal{R}_i^{(t)}\}_i\}_t) = \sum_{i,t} \mathcal{R}_i^{(t)}$.

Equivariant MPNNs are widely used for learning properties of 3D point clouds, such as molecules and materials. However, there are several limitations to their expressiveness,

- **Non-local :** MPNNs are restricted to a finite range, dictated by the number of iterations $T$. A large number of properties in physics are long-range, which requires $T \to \infty$. Adding a large number of layers leads to high computational overhead and poor expressivity due to oversmoothness (Di Giovanni et al., 2023).

- **Correlation order:** Most MPNNs use two body messages, which means that the pooling in Equation 5 is just a sum. The MACE architecture (Batatia et al., 2022b) extended the message construction to an arbitrary order, but increasing the order is still computationally demanding, especially for large $T$.

## 4 MATRIX FUNCTION NEURAL NETWORKS

### 4.1 THE DESIGN SPACE OF MATRIX FUNCTION NEURAL NETWORKS

Our MFN models act in the space of group equivariant matrix operators of the graph, that we denote $\mathcal{H}(\mathcal{G})^G$, where most commonly $G = O(3)$ the rotations group (including in our examples), but our framework applies to the more general class of reductive Lie groups (e.g., the Lorentz group in high energy physics or $SU(3)$ in quantum chromodynamics). In the same setting as in section 3.2, let $\mathcal{G}$ be an undirected graph with $n$ nodes, and let the states $\sigma_i$ be embedded in a configuration space $\Omega$. The architecture operates in four stages at each layer, a **local graph layer** that learns node features from the states, the **matrix construction**, the **matrix function**, and the **update**. These are visually outlined in figure 1.

**Local graph layer**  For each iteration $t$, the first step in an MFN layer is to form equivariant node features using a local graph layer (abstractly noted $\mathcal{L}$) as a function of the **local** environment of a node $i$,

$$\mathcal{L}^{(t)} : \{\sigma_j^{(t)}\}_{j \in \mathcal{N}(i)} \mapsto V_{icm}^{(t)},$$

(6)

where $V_{icm}^{(t)}$ are the learned equivariant node features and $m$ indexes the representation of $G$ acting on $\mathbf{V}$, i.e. $\mathbf{V}_m \circ g = \sum_{mm'} \rho_{mm'}(g)\mathbf{V}_{m'}$. We denote the dimension of the representation by $M$ and the channel dimension by $c$. In the case where $G = O(3)$, the $m$ indices correspond to the usual angular momentum. The specific choice of the graph layer $\mathcal{L}$ depends on the application. One could use any equivariant architecture to learn these node features, such as a layer of equivariant MPNNs. In the case of the rotation group, any such function can be approximated with arbitrary accuracy using an ACE or MACE layer (Zhang et al., 2022; Batatia et al., 2022b). This result has been generalized to any reductive Lie group, using the $G$-MACE architecture (Batatia et al., 2023). It is important to note that the use of more expressive approximators will result in better coverage of the operator space $\mathcal{H}(\mathcal{G})^G$ and therefore better general expressivity.

**Matrix construction**   The second step involves constructing a set of graph matrix operators from the learned node features. The space of graph matrix operators, $\mathcal{H}(\mathcal{G})^G$, corresponds to the space of operators that are (1) **self-adjoint**, (2) **permutation equivariant**, and (3) **G-equivariant**. The entries of the matrix operator correspond to learnable equivariant edge features expanded in the product of two representations $m_1$ and $m_2$. Hence, for each channel $c$, the equivariant operators are square matrices $\mathbf{H}_c \in \mathbb{C}^{Mn \times Mn}$, consistent with $n \times n$ blocks of size $M \times M$, where $n$ are the number of nodes. Each nonzero block $\mathbf{H}_{cij}$ corresponds to learnable edge features between the nodes $i$ and $j$, such that

$$H_{cij,m_1m_2}^{(t)} = \phi_{cm_1m_2}^{(t)}(\mathbf{V}_i^{(t)}, \mathbf{V}_j^{(t)}), \quad \text{if } j \in \mathcal{N}(i) \text{ and } i \in \mathcal{N}(j), \tag{7}$$

where $i, j$ denote the indices of the matrix blocks, while $m_1, m_2$ represent the indices within each block. Additionally, $\phi_{cm_1m_2}^{(t)}$ is a learnable equivariant function mapping a tuple of node features to an edge feature. For a concrete example of matrix construction in the case of isometry group $O(3)$, see section 4.3 and see Figure 2 for an illustration of the block structure.

The full matrix inherits the equivariance of node features,

$$\mathbf{H} \circ g = \boldsymbol{\rho}(g)\mathbf{H}\boldsymbol{\rho}^*(g), \quad \forall g \in G, \tag{8}$$

where $\boldsymbol{\rho}$ is a block diagonal unitary matrix that denotes the group action of $G$ on the basis element $\phi$. In the case where $G$ is the rotation group, the blocks of $\boldsymbol{\rho}$ correspond to the usual $D$-Wigner matrices.

**Matrix function**   The central operation of the MFN architecture is the matrix function, which introduces long-range many-body effects. Any continuous function $f_\theta : \mathbb{R} \to \mathbb{R}$, with parameters $\theta$, can be used to define a matrix function that maps a square matrix to another square matrix of equal size. Formally, a matrix function on self-adjoint matrices $\mathbf{H}$ can be defined by its spectral decomposition. Using the fact that $\mathbf{H}$ is diagonalizable such that $\mathbf{H} = \mathbf{U}\boldsymbol{\Lambda}\mathbf{U}^{\mathbf{T}}$ with $\mathbf{U}$ orthogonal and $\boldsymbol{\Lambda}$ diagonal,

$$f_\theta(\mathbf{H}) = \mathbf{U}f_\theta(\boldsymbol{\Lambda})\mathbf{U}^{\mathbf{T}}, \tag{9}$$

where $f(\boldsymbol{\Lambda})$ is a diagonal matrix obtained by applying $f$ to the each diagonal element of $\boldsymbol{\Lambda}$. An essential observation is that **any** continuous matrix function $f_\theta$ preserves equivariance (see proof in Appendix A.6),

$$f_\theta(\mathbf{H} \circ g) = \boldsymbol{\rho}(g)f_\theta(\mathbf{H})\boldsymbol{\rho}^*(g), \quad \forall g \in G. \tag{10}$$

The matrix function can be related to a high-order many-body equivariant function via the Cayley–Hamilton theorem. The eigendecomposition in Equation 9 is responsible for the non-locality of our approach. In practice, computing matrix functions is expensive as it requires diagonalization, scaling as $n^3$ with the number of nodes. Many approaches are available to approximate matrix functions, such as Chebyshev polynomials or rational approximation, which can leverage potentially cheaper evaluation schemes. Furthermore, the matrix $\mathbf{H}$ is sparse in many applications, which can be further exploited to reduce computational cost. To further optimize this, we propose to employ a resolvent expansion to parameterize $f_\theta$, detailed in Section 4.2. Similar approaches have been successfully applied to large matrices in other fields such as electronic structure calculations (Lin et al., 2009; 2013).

**Update**   The **diagonal update** updates the state of each node with non-local features extracted from the diagonal blocks of the matrix function,

$$h_{icm}^{(t+1)} = V_{icm}^{(t)} + \sum_{\tilde{c}} w_{c\tilde{c}}^{(t)} f_\theta^{(t)}(\mathbf{H}_{\tilde{c}}^{(t)})_{ii,m0} \tag{11}$$

This method is the most computationally efficient since selected inversion techniques (Lin et al., 2009) can be employed to efficiently evaluate the diagonal blocks of a matrix function; cf. Section 4.2. Note that the diagonal blocks are symmetric and therefore extracting $m0$ or $0m$ is equivalent. Alternative updates can be defined from the matrix function that we detail in the appendix A.5. The optimal kind of update is a trade-off between expressivity and computational cost. All of the updates differ fundamentally from the standard spectral GNN update, which is a filtering operation.

The node states are then updated using these new nonlocal node features $\sigma_i^{(t+1)} = (x_i, \boldsymbol{h}_i^{(t+1)})$ to form the next states. The steps are repeated for $T$ iterations, starting from the local graph layer.

**Readout** The readout phase is the same as the usual MPNN readout in Equation 5.

## 4.2 RESOLVENT PARAMETERIZATION OF MATRIX FUNCTIONS

The evaluation of the matrix function in Equation 9 is the practical bottleneck of our method. The cost of the evaluation depends on the choice of parameterization of the univariate function $f_\theta$. For a general analytic $\tilde{f} : \mathbb{C} \to \mathbb{C}$, resolvent calculus allows us to represent

$$\tilde{f}(\mathbf{H}) = \oint_{\mathcal{C}} \tilde{f}(z)(z\mathbf{I} - \mathbf{H})^{-1} \frac{dz}{2\pi i},\tag{12}$$

where $\mathcal{C}$ is a curve encircling the eigenvalues of $\mathbf{H}$ and excluding any poles of $\tilde{f}$. Approximating the contour integration with a quadrature rule with nodes $z_s$ and weights $\tilde{w}_s$ yields $\tilde{f}(\mathbf{H}) \approx \sum_s \tilde{w}_s \tilde{f}(z_s)(z_s\mathbf{I} - \mathbf{H})^{-1}$, and merging $w_s := \tilde{w}_s \tilde{f}(z_s)$ we arrive at the parameterization

$$f_\theta(\mathbf{H}) = \sum_s w_s(z_s\mathbf{I} - \mathbf{H})^{-1}.\tag{13}$$

Pole expansions for evaluating matrix functions have a long and successful history, especially when the arguments are sparse matrices (Higham, 2008). The novelty in equation 13 over standard usage of pole expansions in computing matrix functions (Higham, 2008) is that both the *weights* $w_s$ and the *poles* $z_s$ are now learnable parameters.

The derivation shows that in the limit of infinitely many pole-weight pairs $(z_s, w_s)$ any analytic matrix function can be represented. Since analytic functions are dense in the space of continuous functions, this means that all continuous matrix functions can be represented in that limit as well (at the same time letting the poles approach the spectrum). In practice, the poles should be chosen with non-zero Imaginary parts in order to avoid the spectrum of $\mathbf{H}$, which is real since $\mathbf{H}$ is assumed to be self-adjoint. Therefore, we choose adjoint pole weight pairs $(w_s, z_s)$ and $(w_s^*, z_s^*)$ to ensure that $f_\theta$ is real when restricted to real arguments.

**Linear scaling cost** The pole expansion framework is the first key ingredient in linear scaling electronic structure methods Goedecker (1999) such as PEXSI Lin et al. (2009; 2013). The second ingredient is the selected inversion of the resolvent. Instead of computing the full inverse, $(z\mathbf{I} - \mathbf{H})^{-1}$, one first computes a sparse $LDL^*$ factorization and then selectively computes only the diagonal entries of the resolvent. The bottleneck in this approach is the $LDL^*$ factorization. For a full matrix, it scales as $O(n^3)$ operations and $O(n^2)$ memory. The complexity improves considerably for sparse matrices. Suppose that the sparsity pattern is $d$-dimensional, corresponding to a topologically $d$-dimensional graph; e.g. the cumulenes in Section 5.1 are topologically one-dimensional despite being embedded in $\mathbb{R}^3$. Using nested dissection ordering to minimize the fill-in, the cost of the $LDL^*$ factorization reduces to $O(n)$ for $d = 1$ (e.g., natural language processing and quasi-1D molecular systems such as carbon-nano-tubes); $O(n^{3/2})$ operations and $O(n \log n)$ memory for $d = 2$ (e.g., Image recognition); and $O(n^2)$ operations and $O(n^{4/3})$ memory for $d = 3$. A more formal exposition of this material is given in Davis (2006). The final step to reduce the computational cost of our architecture to *linear scaling* is to replace the $LDL^*$ factorization with an incomplete factorization, as proposed by Etter (2020) in the context of electronic structure calculations. This would lead to approximations in non-locality that need to be investigated in future work.

## 4.3 EXAMPLE OF MFN FOR THE $O(3)$-GROUP

Here we detail a concrete example for the MFN layer in the case of $O(3)$-group. For concreteness, we will consider point clouds of atoms as input. We will reuse this architecture in the Experiment

section. At each layer $t$, the states of each node are described by a tuple $\sigma_i = (\mathbf{r}_i, \theta_i, h_i^{(t)})$, where $\mathbf{r}_i \in \mathbb{R}^3$ denotes the position of atoms in the 3D space, $\theta_i \in \mathbb{Z}$ is the nuclear charge and $h_i^{(t)}$ the learnable node features that the MFN layer will update that we initialize just as a learnable embedding of the species. The node features are expanded on the spherical basis and use the usual convention of the $\tilde{l}\tilde{m}$ index (with the correspondence with the single $m$ in Section 4.1 being $\big(m = 0 \to (\tilde{l} = 0, \tilde{m} = 0)$ and $m = 1 \to (\tilde{l} = 1, \tilde{m} = -1)$ etc.$\big)$. For the **local graph layer**, we use a MACE layer (see Batatia et al. (2022b) for details) that constructs the local node features $V_{i,c\tilde{l}\tilde{m}}$ in Equation 6. We then construct the matrix using a tensor product of the features of the nodes on nodes $i$ and $j$,

$$H_{cij,\tilde{l}_1\tilde{m}_1\tilde{l}_2\tilde{m}_2} = R_c(r_{ij})(V_{i,c\tilde{l}_1\tilde{m}_1} \otimes V_{j,c\tilde{l}_2\tilde{m}_2}), \quad \text{if } j \in \mathcal{N}(i) \text{ and } i \in \mathcal{N}(j), \tag{14}$$

where $R$ corresponds to a function of the distance $r_{ij}$ between $i$ and $j$. As the graph is undirected, the matrix is symmetric with respect to $i, j$. The matrix is sparse, as only the elements corresponding to

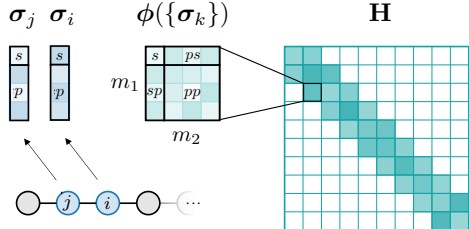

Figure 2: **Block structure** of a Euclidean MFN operator, $\mathbf{H}$. Each entry in $\mathbf{H}$ corresponds to a different product of representations of the group of rotations $(ss, sp, pp, ...)$. Example for L=1.

an edge of the graph are nonzero. This matrix exhibits a block structure illustrated in Fig. 2 similar to Hamiltonian matrices in quantum mechanics with $ss$ and $sp$ orbitials. We refer to $L$ as the maximal value of each spherical index $\tilde{l}_1$ and $\tilde{l}_2$ of the matrix $(0 \leq \tilde{l}_{1,2} \leq L)$, and we refer to this architecture as $\text{MFN}_L^{(\text{MACE})}$ in the Results section.

### 4.4 EXPRESSIVITY OF MATRIX FUNCTION NETWORKS

Purely formally, one may think of a matrix function $\tilde{f}(\mathbf{H})$ as an infinite power series. This suggests that MFNs are inherently non-local and exhibit a convolution-like structure, similar to message-passing methods with infinite layers and linear update (see Appendix A.2 for details). This is of interest for modeling causal relationships or non-local interactions by proxy, such as in chemistry or natural language processing. In these cases, the propagation of local effects over long distances results in multiscale effects that are effectively captured by our method. The degree of non-locality of the interaction in MFN can be precisely quantified using the Combes-Thomas theorem (Combes & Thomas, 1973). We provide a detail analysis in the Appendix A.1.

## 5 RESULTS

### 5.1 CUMULENES: NON-LOCAL 3D GRAPHS

We compare the non-locality of MFNs to local MPNNs and global attention MPNNs using linear carbon chains, called cumulenes. This example is a notorious challenge for state-of-the-art architectures machine learning force fields Unke et al. (2021c), and our main motivation for introducing the MFN architecture.

Cumulenes are made up of double-bonded carbon atoms terminated with two hydrogen atoms at each end. Cumulenes exhibit pronounced non-local behavior as a result of strong electron delocalization. Small changes in chain length and relative angle between the terminating hydrogen atoms can result in large changes in the energy of the system, as visually represented in Figure 3. These structures are known to illustrate the limited expressivity of local models (Unke et al., 2021c) and are similar to the k-chains introduced by Joshi et al. (2023) in the context of the geometric WL test. Frank et al. (2022) showed that global attention is capable of capturing the angular trends of cumulenes with fixed length. We go beyond and demonstrate that MFNs are capable of accurately

extrapolating to longer chains, simultaneously capturing length and angle trends. In contrast, global attention models such as Spookynet (Unke et al., 2021b), are unable to extrapolate to longer chains, highlighting the benefit of the matrix function formalism. For all MFN models in this section, we use MACE layers (Batatia et al., 2022b) to form the matrix at each layer. We refer to the model as MFN (MACE). Details of the training set and the specific choice of parameterization of the matrix entries are included in the Appendix A.9.

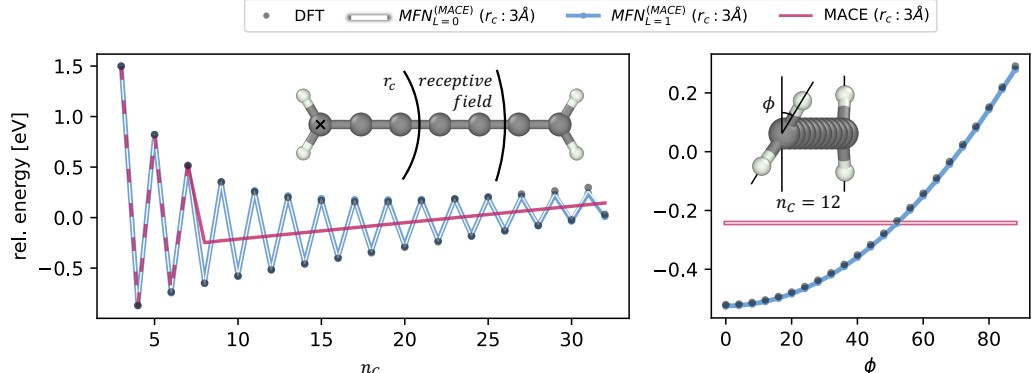

Figure 3: **Visualizing MFN expressivity on cumulene chains.** The left panel depicts energy trends with respect to cumulene chain length at a fixed angle $\phi = 5°$. The right panel shows the DFT (ground truth) and the predicted energy as a function of the dihedral angle $\phi$ between the hydrogen atoms for a cumulene chain containing 12 carbon atoms. Local many-body equivariant models (MACE) are only able to capture average trends, even though test configurations are included in the training set. Invariant MFNs ($L = 0$) capture only the trends with respect to length, while equivariant MFNs ($L = 1$) capture both non-local trends. All models have a cutoff distance $r_c$ of 3Å, corresponding to the nearest neighbors, with two message-passing layers. The cutoff distance as well as MACE's receptive field for the first carbon atom is annotated in the left panel.

**Trends with chain length and rotation** The lowest energy structure of cumulenes alternates between 90- and 0-degree angles for odd and even carbon atom counts, respectively. Consequently, varying the number of carbon atoms at a fixed angle unveils a distinctive zigzag pattern in the ground truth energy (Fig. 3 left panel). Although the local model, with two message passing layers, is trained on exactly these configurations, this system is beyond its expressivity, as can be seen by the linear trend for $n_c > 7$ in (Figure 3 left panel). In contrast, the invariant and equivariant MFN models perfectly reproduce density functional theory (DFT) energies, thanks to their inherent non-locality. To demonstrate the necessity of equivariance, we train models on the energy of a fixed size cumulene as a function of the dihedral angle between the hydrogen atoms. Figure 3 demonstrates that only the equivariant MFN (L=1) captures the angular trend.

**Guaranteed Non-Local dataset** Datasets designed to test non-local effects often yield unexpectedly high accuracy when evaluated with local models (Kovacs et al., 2023), complicating the assessment of model non-locality. The dataset introduced here is based on cumulenes, whose strong electronic delocalization results in a directly observable non-locality. The training set contains geometry-optimized cumulenes with 3-10 and 13, 14 carbon atoms, which are then rattled and rotated at various angles. The test set contains cumulenes created in a similar fashion with the same number of carbons (in-domain) and cumulenes of unseen length, not present in the dataset (out-domain 11,12 and 15,16). Table 1 shows that the MFN architecture significantly outperforms both local and attention-based models (Spookynet). Attention captures some non-locality, resulting in marginally lower errors on the train and in-domain test set. However, the learned non-locality does not generalize to larger molecules, obtaining energy and forces worse than those obtained with a simple local model. The structured non-locality of MFNs enables generalization to larger unseen system sizes.

## 5.2 PERFORMANCE ON PURE GRAPHS

In this section, we evaluate the performance of our MFNs models in graph-level prediction tasks using GCN layers for the matrix construction. Detailed of the datasets and hyperparameters of the models can be found in the Appendix A.9.

| Dataset | $n_C$ | E (meV/atom) | | | F (meV/A) | | |
|---|---|---|---|---|---|---|---|
| | | MACE (Local) | SpookyNet (Global attention) | MFN$^{(MACE)}$ (ours) | MACE (Local) | SpookyNet (Global attention) | MFN$^{(MACE)}$ (ours) |
| Train | 3-10,13,14 | 41.1 | 31.4 | **2.0** | 179.6 | 114.1 | **31.7** |
| Test (In Domain) | 3-10,13,14 | 41.8 | 30.8 | **2.6** | 205.6 | 162.3 | **34.0** |
| Test (Out Domain) | 11,12 | 16.5 | 31.4 | **0.9** | 108.5 | 116.2 | **22.5** |
| Test (Out Domain) | 15,16 | 12.0 | 23.4 | **2.6** | 77.1 | 87.6 | **37.7** |

Table 1: **Guaranteed non-local cumulene dataset** containing rattled cumulene chains, with various chain lengths ($n_C$) and hydrogen dihedral angles ($\phi$). The table compares energy (E) and forces (F) RMSEs between local two-layer MPNNs (MACE), global attention MPNNs (SpookyNet), and equivariant MFNs. Train and in-domain test sets contain cumulenes of lengths 3-10 and 13,14. Two out-domain test sets compare different levels of extrapolation to unseen cumulene lengths, containing cumulenes with 11, 12 and 15, 16 carbon atoms, respectively. Bold is best and underline second best.

**ZINC.** We use the default dataset splits for ZINC, aligned with the leaderboard baseline settings, with approximately 500K parameters set. Table 2 shows that MFN surpasses previous architectures, demonstrating the utility of learning various operators, even on pure graphs.

Table 2: Results on ZINC with the MAE and number of parameters used, where the best results are in bold. Baselines are taken from (Yang et al., 2023) and model citations are in A.9.2.

| Method | GCN | GAT | MPNN | GT | SAN | Graphormer | PDF | MFN (GCN) |
|---|---|---|---|---|---|---|---|---|
| MAE | 0.367±0.011 | 0.384±0.007 | 0.145±0.007 | 0.226±0.014 | 0.139±0.006 | 0.122±0.006 | 0.066±0.004 | **0.063±0.002** |
| #para | 505k | 531k | 481k | NA | 509k | 489k | 472k | 512k |

**TU Datasests.** We test our model on five TUDataset datasets involving both bioinformatics datasets (MUTAG, ENZYMES, PTC MR, and PROTEINS) and a social network dataset (IMDB-B). To ensure a fair comparison with baselines, we follow the standard 10-fold cross-validation and dataset split in table 3.

Table 3: Results on TUDataset (Higher is better). Bold is best, and underlined second best within $\pm0.5\%$. Baselines are taken from (Yang et al., 2023) and model citations are in A.9.3

| Method | MUTAG | ENZYMES | PTC_MR | PROTEINS | IMDB-B |
|---|---|---|---|---|---|
| GK | 81.52±2.11 | 32.70±1.20 | 55.65±0.5 | 71.39±0.3 | - |
| RW | 79.11±2.1 | 24.16±1.64 | 55.91±0.3 | 59.57±0.1 | - |
| PK | 76.0±2.7 | - | 59.5±2.4 | 73.68±0.7 | - |
| AWE | 87.87±9.76 | 35.77±5.93 | - | - | 74.45±5.80 |
| PSCN | 88.95±4.4 | - | 62.29±5.7 | 75±2.5 | 71±2.3 |
| ECC | 76.11 | 45.67 | - | - | - |
| DGK | 87.44±2.72 | 53.43±0.91 | 60.08±2.6 | 75.68±0.5 | 66.96±0.6 |
| GraphSAGE | 85.1±7.6 | 58.2±6.0 | - | - | 72.3±5.3 |
| CapsGNN | 88.67±6.88 | 54.67±5.67 | - | 76.2±3.6 | 73.1±4.8 |
| GIN | 89.4±5.6 | - | 64.6±7.0 | 76.2±2.8 | 75.1±5.1 |
| $k$-GNN | 86.1 | - | 60.9 | 75.5 | 74.2 |
| IGN | 83.89±12.95 | - | 58.53±6.86 | 76.58±5.49 | 72.0±5.54 |
| PPGNN | 90.55±8.7 | - | 66.17±6.54 | **77.20±4.73** | 73.0±5.77 |
| GCN$^2$ | 89.39±1.60 | - | 66.84±1.79 | 71.71±1.04 | 74.80±2.01 |
| PDF | 89.91±4.35 | **73.50±6.39** | 68.36±8.38 | 76.28±5.1 | **75.60±2.69** |
| MFN (GCN) | **91.5±7.35** | 72.9±7.55 | **68.9±8.09** | 76.18±4.07 | 74.1±1.04 |

## 6 CONCLUSION

We have introduced Matrix Function Networks (MFNs), an architecture designed to address the limitations of existing GNNs and MPNNs in modeling non-local many-body interactions. Utilizing a resolvent expansion, MFNs achieve potentially linear scaling with respect to system size, offering a computationally efficient solution. Our evaluations indicate state-of-the-art performance on ZINC and TU graph datasets without human-designed features to capture the global graph topology. We also demonstrate that our architecture is capable of modeling the complex non-local interactions of cumulene quantum systems. Future work could focus on extending MFNs to other complex systems, further validating its adaptability and efficiency, and exploring its interpretability.

## ACKNOWLEDGMENTS

CO's work was supported by the NSERC Discovery Grant IDGR019381 and the NFRF Exploration Grant GR022937. LLS acknowledges support from the EPSRC Syntech CDT with grant reference EP/S024220/1. Computational resources were provided by the Cambridge Service for Data Driven Discovery (CSD3), which was accessed through the University of Cambridge EPSRC Core Equipment Award EP/X034712/1. The authors would also like to thank Rokas Elijošius for useful discussions.

## ETHICS STATEMENT

FAF is employed by AstraZeneca at time of publication; however, none of the work presented in this manuscript was conducted at or influenced by this affiliation.

## REPRODUCIBILITY STATEMENT

To ensure reproducibility and completeness, we include detailed descriptions of the model used, hyperparameters, and data sets in the Appendix. The ZINC and TU datasets are publicly available. The cumulene dataset is available at: `https://github.com/LarsSchaaf/Guaranteed-Non-Local-Molecular-Dataset`. The MFN code will be available at: `https://github.com/ilyes319/mfn`.

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

# A APPENDIX

## A.1 NON-LOCALITY OF MFNS AND COMBES-THOMAS THEOREM

The degree of non-locality of the interaction can be precisely quantified. If $\mathbf{H}$ has a finite interaction range, then the Combes-Thomas theorem Combes & Thomas (1973) implies that $\left|(z\mathbf{I} - \mathbf{H})^{-1}_{i,j}\right| \leq Ce^{-c\gamma_z d_{ij}}$, where $\gamma_z = \mathrm{dist}(z, \sigma(\mathbf{H}))$, $d_{ij}$ is the length of the shortest path from node $i$ to node $j$, and $C, c$ may depend on the norm of $\mathbf{H}$ but can be chosen uniformly provided $\|\mathbf{H}\|_\infty$ (operator max-norm) has an upper bound. Our normalisation of weights in Appendix A.4 is an even stronger requirement and hence guarantees this.

Since we have taken $\mathbf{H}$ self-adjoint with real spectrum, an estimate for $\gamma_z$ is $\mathrm{Imag}(z)$. As a result, if we constrain the poles in the parameterization of $f_\theta$ to be at $\mathrm{Imag}(z_s) = \gamma$, then the resulting matrix function will satisfy

$$\left|[f_\theta(\mathbf{H})]_{i,j}\right| \leq Ce^{-c\gamma d_{ij}}. \tag{15}$$

Therefore, the degree of locality can be controlled by constraining $\mathrm{Imag}(z_s)$ to be some fixed value. While equation 15 only gives an upper bound, it is not difficult to see that it can in fact be attained for a suitable choice of matrix function. In practice, however, $f_\theta$ can be seen as approximating an unknown target $\tilde{f}$ with learnable weights $w_s$. If $\tilde{f}$ is analytic in a neighborhood of $\sigma(\mathbf{H})$, then $\left|[\tilde{f}(\mathbf{H})]_{i,j}\right| \leq Ce^{-c\tilde{\gamma}d_{ij}}$, where $\tilde{\gamma}$ measures how smooth $\tilde{f}$ is (distance of poles of $\tilde{f}$ to $\sigma(\mathbf{H})$). For our parameterization $f_\theta$ we therefore expect the same behavior $\gamma = \tilde{\gamma}$ in the pre-asymptotic regime of moderate $d_{ij}$ but transition to the asymptotic rate equation 15 for large $d_{ij}$. An in-depth analysis of such effects go beyond the scope of this work.

## A.2 GEOMETRIC EXPRESSIVENESS AND RELATIONSHIP TO INFINITE LAYERS OF MPNNS

The WL-test quantifies a network's ability to distinguish non-isomorphic graphs. Its extension to graphs embedded in vector spaces is known as the geometric WL-test (Joshi et al., 2023). The expressivity in the following discussion adheres to these definitions. In the context of graphs in $\mathbb{R}^3$, practical relevance is high. A clear relationship exists between equivariant MPNNs with linear updates and infinite layers and a one-layer MFN due to the matrix function's product structure. The features constructed by a one-layer MFN with a single matrix channel and two-body matrix entries closely resemble the features of a two-body equivariant MPNN with **infinite layers** and **linear updates**. When matrix entries incorporate features beyond two-body, the one-layer MFN becomes more expressive than its two-body feature counterpart.

Here, we give an intuition of the relationship between the two approaches. Any analytic matrix function $f$ admits a formal power series expansion, valid in its radius of analyticity,

$$f(H) = \sum_{k=0}^{\infty} c_k H^k \tag{16}$$

where $c_k$ are the complex (or real) coefficients of the expansion. Let us look at the diagonal elements of each of these powers that we will extract as the next doe features,

$$(H)^2_{ii,c00} = \sum_{j \in \mathcal{N}(i)} H_{ij,c0m} H_{ji,cm0}, \quad (H)^3_{ii,c00} = \sum_{j \in \mathcal{N}(i)} \sum_{k \in \mathcal{N}(j)} H_{ij,c0m} H_{jk,cmm'} H_{ji,cm'0} \tag{17}$$

The sum over $\mathcal{N}(i)$ is forced due to the sparisity of the constructed matrix $H$ in our method. Therefore, we observe a convolutional structure similar to message-passing methods. The power 2, corresponds to a single convolution, the power 3 to two convolutions, all the way to infinity. Each coefficient of the matrix function that is **learnable** can act like weights in the linear update of the analogous layer. Because the power series expansion is infinite, one sees a clear analogy between the two.

## A.3 INTERPRETABILITY OF THE MFN OPERATORS

In the case of the Euclidean group, the matrices learned in MFNs have the same symmetries as Euclidean operators. Euclidean operators play a crucial role in quantum mechanics. They are defined as self-adjoint operators (in the space of square-integrable functions) that are equivariant to the action of the group of rotations, translations, reflections, and permutations. When these operators are expended on the basis of the irreducible representations of the rotation group, they exhibit a block structure (see Appendix A.6) in which each entry is labeled with representations $(ss, ps, pp, dd, ...)$.

The most central of these operators is the Hamiltonian operator. The energy of a quantum system is related to the trace of a specific matrix function of the Hamiltonian,

$$E = \mathrm{Tr}f(\mathbf{H}) \tag{18}$$

The Hamiltonian is usually computed as a fixed point of a self-consistent loop that minimizes the energy. This loop introduces many-body non-local effects. This motivates us to use many-body functions to parameterize our matrices and to learn the fixed point directly via learning a matrix function of many matrices. This is in opposition to tight-binding methods that usually construct a single low-body Hamiltonian with physics-inspired functional forms but require several self-consistent iterations to converge.

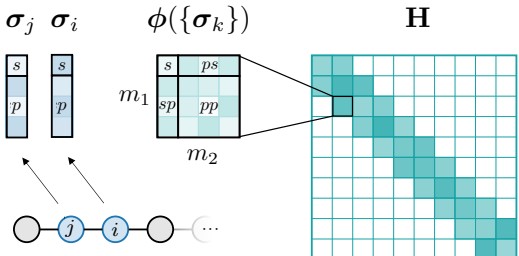

Figure 4: **Block structure** of a Euclidean Operator, H, learnt in the MFNs. Each entry in H corresponds to a different product of representations of the group of rotations $(ss, sp, pp, ...)$.

### A.4 MATRIX NORMALIZATION

Batch normalization (Ioffe & Szegedy, 2015) and layer normalization (Ba et al., 2016) are two techniques widely used in deep learning to stabilize training, improve convergence, and provide some form of regularization.

In the context of MFNs, a similar normalization strategy is needed to ensure stable training and improved convergence. Instead of normalizing the features directly as conventional normalization layers do, we normalize the eigenvalues of a set of H matrices with batch dimension $[$ batch, channels, $n$, $n]$, where $n$ is the size of the matrix. The aim is to adjust their mean and variance to 0 and 1, respectively, much like standardization in traditional batch or layer normalization. For the batch matrix norm, the mean is taken across the batch dimension, and for the layer matrix norm, the normalization is taken across the channel dimension. The mean and the variance of the eigenvalues are computed using the following formulas,

$$\mathbb{E}(\Lambda) = \frac{\text{tr}(\mathbf{H})}{n}, \quad \text{Var}(\Lambda) = \frac{\text{tr}(\mathbf{H}^2)}{n-1} - \frac{\text{tr}(\mathbf{H})^2}{n(n-1)} \tag{19}$$

This normalization of eigenvalues ensures that the spectral properties of the graph used for representation in the MFN are effectively standardized, contributing to better training stability and convergence.

### A.5 ALTERNATIVE UPDATES FOR MFN

Many updates can be defined in order to extract non-local features from the matrix function.

1. The **dense update** utilizes the entire matrix, including all **off-diagonal terms**, to update the next matrix function,

$$h_{icm}^{(t+1)} = V_{icm}^{(t)} + \sum_{\tilde{c}} w_{c\tilde{c}}^{(t)} f_\theta^{(t)} (\mathbf{H}_{\tilde{c}}^{(t)} + f_\theta^{(t-1)} (\mathbf{H}_{\tilde{c}}^{(t-1)}))_{ii,m0} \tag{20}$$

This update can not be performed using selected inversion techniques, but it can offer additional expressiveness guarantees.

2. The **sparse update:** uses only the parts of the matrix corresponding to the connected nodes to update the nodes and edges of the graph to obtain the matrix function in the next layer,

$$h_{icm}^{(t+1)} = V_{icm}^{(t)} + \sum_{\tilde{c}} w_{c\tilde{c}}^{(t)} f_\theta^{(t)} (\mathbf{H}_{\tilde{c}}^{(t)} + \{f_\theta^{(t-1)} (\mathbf{H}_{\tilde{c}}^{(t-1)})_{ij}\}_{j \in \mathcal{N}(i)})_{ii,m0}. \tag{21}$$

### A.6 EQUIVARIANCE

**Proof of equivariance of continuous matrix functions**   We will first prove the invariance of the spectrum of an operator. Let $H \in \mathbb{C}^{n \times n}$ be a self-adjoint matrix. $H$ admits an eigendecomposition, $H = U\Lambda U^T$. The eigenvalues $\Lambda$ are the roots of the characteristic polynomial of $H$,

$$\det(H - \lambda I_n) = 0, \quad \forall \lambda \in \Lambda \tag{22}$$

Let $G$ be a reductive Lie group. Then any finite-dimensional representation of $G$, can be viewed as a representation of the maximal compact subgroup of the complexification of $G$, by the Weyl unitary trick. Therefore, they can analytically continued to unitary representations. Let $\rho : G \rightarrow \mathbb{C}^{n \times n}$ be a representation of $G$ taken unitary, $\rho(g)\rho(g)^* = I_n, \forall g \in G$. Let $H_1$ and $H_2$ be two matrices related by an action of $G$, then $H_1 = \rho(g)H_2\rho(g)^*$ for some $g \in G$. Let $\Lambda_1$ and $\Lambda_2$ be their respective spectrums. For all $\lambda \in \Lambda_1$

$$\det(H_1 - \lambda I_n) = \det(\rho(g)H_2\rho(g)^* - \lambda I_n) = \det(\rho(g)(H_2 - \lambda I_n)\rho(g)^*) \tag{23}$$

$$= det(\rho(g)\rho(g)^*)det(H_2 - \lambda I_n) = det(H_2 - \lambda I_n) = 0 \tag{24}$$

Therefore the eigenvalues of $H_1$ are also eigenvalues of $H_2$ and $\Lambda_1 = \Lambda_2$.

We will now prove that if $u \in U_1$ is an eigenvector of $H_1$ with eigenvalue $\lambda$ then $\rho(g)u$ is an eigenvector of $H_2$. For all $u \in U_1$

$$H_2(\rho(g)u) = \rho(g)H_1\rho(g)^*(\rho(g)u) = \rho(g)H_1 u \tag{25}$$

$$= \lambda(\rho(g)u) \tag{26}$$

Now let's prove the equivariance of matrix function. Let $f : \mathbb{C} \to \mathbb{C}$ be a continuous function, then a matrix function is defined as,

$$f(H) = U f(\Lambda) U^T \tag{27}$$

Therefore for any $g \in G$,

$$f(H \circ g) = f(\rho(g)H\rho(g)^*) = \rho(g)U f(\Lambda) U^T \rho(g)^* \tag{28}$$

$$= \rho(g)f(H)\rho(g)^* \tag{29}$$

$$\square$$

**Equivariance of the resolvent** The pole expansion yields a straightforward proof of $f_\theta(\mathbf{H})$ equivariance:

$$f_\theta(\mathbf{H} \circ g) = \sum_s w_s(z_s\mathbf{I} - \boldsymbol{\rho}\mathbf{H}\boldsymbol{\rho}^*)^{-1} = \sum_s w_s\boldsymbol{\rho}(z_s\mathbf{I} - \mathbf{H})^{-1}\boldsymbol{\rho}^* = \boldsymbol{\rho}f_\theta(\mathbf{H})\boldsymbol{\rho}^*. \tag{30}$$

For a general continuous function $\tilde{f}$, the analogous result follows by density.

## A.7 MULTIVARIATE MATRIX FUNCTION

As MFNs extract features from multiple operators, it is useful to extend the univariate matrix function in Equation 9 to a multivariate matrix function. The space of multivariate functions $\mathcal{F}(\mathcal{H}(\mathcal{G}) \times ... \times \mathcal{H}(\mathcal{G}))$ is isomorphic to the closure of the span of the tensor product of functions of single operators $\mathcal{F}(\mathcal{H}(\mathcal{G}))^{\otimes n}$, We call the number of variables $n$, the correlation order of the matrix function. The resolvent expansion can be generalized to the multivariate matrix function, using matrix products of resolvents,

$$f(\mathbf{H}_1, ..., \mathbf{H}_n) = \frac{1}{(2\pi i)^n} \int \cdots \iint_{\partial D_1 \times \cdots \times \partial D_n} \frac{f(z_1, \ldots, z_n)}{(z_1\mathbf{I} - \mathbf{H}_1)\cdots(z_n\mathbf{I} - \mathbf{H}_n)} dz_1 \cdots dz_n \tag{31}$$

Multivariate matrix functions create higher order features. A one-layer MFN with a matrix function of correlation order $n$ is as expressive as a $(n + 1)$-**body** equivariant MPNN with **infinite layers** and **linear updates**. The space $\mathcal{F}(\mathcal{H}(\mathcal{G}))^{\otimes n}$ is of very high dimension and it is possible to use standard compression techniques to find more efficient approximations such as tensor decomposition (Darby et al., 2023). The use of linear combinations of matrices in Equation 7 approximates some of this space. We leave further exploration of these kinds of network for future work.

## A.8 RUNTIME COMPARISON

The current MFN implementation is a prototype implementation and has a cubic scaling. However, fast inversion methods are well established, and we explain in Section 4.2 how this scaling can be reduced significantly to less than quadratic scaling, all the way to linear scaling, by exploiting the sparsity of the matrices we construct and selected inversion methods.

Although these methods are known, they require significant effort to implement and integrate them into machine learning libraries like Pytorch due to the need for specialized CUDA kernels. We intend to make this effort and release open-source code in future work, but we believe that this is beyond the scope of this paper, where we focus on the novelty of the architecture and its expressivity.

For the sake of completeness, we show run-time comparison on the cumulene examples between MACE and the MFN. We time the models energy and forces evaluation on an A100 GPU. The graph layer construction of the MFN model is a MACE layer with the same hyperparameters as the MACE model. We use a one layer MFN model with 16 poles and 16 matrix channels. The MACE model has 128 channels, maximal angular resolution of $l_{max} = 3$ and message passing equivariance $L = 1$.

## A.9 DETAILS OF NUMERICAL EXPERIMENTS

### A.9.1 CUMULENES

**Dataset** The cumulene dataset is designed to test the expressivity of graph neural networks and their non-local capabilities. The task is to regress energy and forces from 3D molecular configurations. The ground truth energy is obtained using the ORCA quantum chemistry code at the density functional theory level of accuracy using the *wB97X-D3* functional, the *def2-TZVP* basis set, and very tight SCF convergence criteria.

Table 4: **Runtime comparison** of a energy and force call of a MACE model and MFN model on a 20 atoms cumulene. The MFN models constructs equivariant matrices up to $L = 1$.

| Method | MACE | MFN$^{\text{MACE}}$ |
|---|---|---|
| Time (ms/atom) | 0.7 | 2.8 |

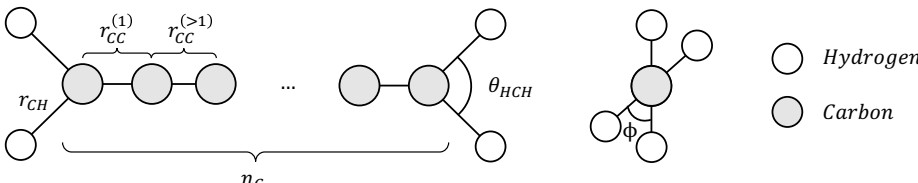

Figure 5: **Angles and distances that define a cumulene graph** used to test expressivity in Figure 3. The carbon-hydrogen ($r_{CH}$), first carbon-carbon ($r_{CC}^{(1)}$), and remaining carbon-carbon distances ($r_{CC}^{(>1)}$) are set to 1.086 Å, 1.315 Å and 1.279 Å respectively. The angle between the hydrogen-carbon-hydrogen ($\theta_{HCH}$) is fixed at 118.71 degrees and the dihedral angle $\phi$ depends on the experiment as detailed in the main text.

**Length and angle scans**  The configurations for the length and angle scans consist of chain-like graphs as in Figure 5. The length scans range from 3 to 30 atoms fixed at an angle of 5 degrees. Additionally, we scan the angle between the two terminating hydrogen groups for a cumulene with twelve carbon atoms ($n_C = 12$). At increments of 3 degrees, we scan from 0 to 90 degrees. By symmetry, this covers the entire range of unique configurations from 0 to 360 degrees. For the expressivity scan, all internal coordinates (see Figure 5) that uniquely define the cumulene are kept constant and set to the geometry-optimised configuration of the length 30 cumulenes. The carbon-hydrogen, first carbon-carbon, and remaining carbon-carbon distances are set to 1.086 Å, 1.315 Å and 1.279 Å respectively (see Figure 5). For the length 12 cumulene, the distance between the most distant carbons is 14.1Å. Thus, it becomes impossible for a purely local equivariant MPNN, with cutoff 3Å and two message-passing layers, to differentiate between two graphs at different angles.

Note that the relative energies of 3 are obtained by subtracting individual atom energies. These are -16.3eV and -1036.1eV for each hydrogen and carbon atoms respectively.

**Guaranteed Non-local dataset**  The GNL dataset tests how well a model can extrapolate to an unseen number of atoms. Furthermore, configurations are rattled to see how well the model can capture both local and non-local effects. Configurations are generated starting from the relaxed geometries of cumulenes with length 0-20. Relaxations are carried out with the smaller *6-31G* basis set. The cumulenes are subsequently rotated by an equally spaced scan with increment 6 degrees, starting from a random angle, and the positions are subsequently randomly perturbed by Gaussian noise with standard deviation 0.01 Å. The training and validation set contain cumulenes of length 0-10 and 13-14, with 10 and 3 samples, respectively. The test set contains in-domain configurations with two samples for each of the lengths present in the training set. Furthermore, it contains configurations with 11-12 and 15-16 carbon atoms, labeled out-of-domain. In total, the train, validation, and test set contain 200, 50 and 170 configurations.

**MACE and MFN Model**  Both the local MACE and MFN are trained on the same graphs, which means that the cutoff distance is fixed at 3Å, including information from the nearest neighbor. For the MFN model, we use the architecture described in Section 4.3. Both models are trained with two layers, allowing the local model to distinguish changes that are separated by 12Å. The MFN is trained using 16 matrix channels and 16 poles. For the matrix construction step, we use a MACE Batatia et al. (2022b) layer with a correlation order 3, $l_{\max} = 3$, $L = 1$ and 128 channels to generate node features $h_i$. We use the diagonal update 11 to update the node features of MACE and reiterate. The readout at the first layer is a linear and at the second it is a one layer MLP with 16 hidden dimensions.

**Spookynet Model**  The Spookynet architecture was trained on the same dataset as the MFN model. We use a one-layer model with 5.5 Å cutoff to reproduce the receptive field of MACE. We use the global attention and electrostatic interactions. The model has 128 channels.

**Training**  All models underwent initial training on the GNL dataset before transfer learning was applied to the relevant length and angle scan data sets. The saved model corresponds to the epoch that exhibits the minimum loss values. Details on settings such as learning rate and epoch count are disclosed in Table 5. Training incorporated both energy and forces, with adjustable weights for each observable. In particular, for

length and angle scans, an additional 100 epochs with zero force weight were used after initial training, ensuring the depiction of the minimal possible energy error in Figure 3, as the lowest-loss model is saved.

Table 5: Model training parameters. For the matrix functions the number of poles ($n_p$) and matrix channels ($c$) are indicated.

| dataset | model | epochs | lr | $E_{\text{weight}}$ | $F_{\text{weight}}$ | $n_{\text{layers}}$ | $r_{\text{max}}$ | other |
|---|---|---|---|---|---|---|---|---|
| **Length Scan** | MFN (L=0) | 1240 | 1e-2 | 1000 | 100 | 2 | 3 | $n_p$=16,$c$=16 |
| | MFN (L=1) | 5628 | 1e-5 | 1000 | 100 | 2 | 3 | $n_p$=16,$c$=16 |
| | MACE | 5288 | 0.005 | 1000 | 100 | 2 | 3 | - |
| **Angle Scan** | MFN (L=0) | 1240 | 1e-2 | 1000 | 100 | 2 | 3 | $n_p$=16,$c$=16 |
| | MFN (L=1) | 656 | 1e-5 | 1000 | 100 | 2 | 3 | $n_p$=16,$c$=16 |
| | MACE | 954 | 0.005 | 1000 | 100 | 2 | 3 | - |
| **GNL** | MFN (L=1) | 4958 | 1e-2 | 100 | 1 | 2 | 3 | $n_p$=16,$c$=16 |
| | MACE | 1260 | 1e-2 | 100 | 1 | 2 | 3 | - |
| | Spookynet | 3500 | 1e-4 | 0.10 | 0.90 | 1 | 5.5 | attention |

## A.9.2 ZINC

**Dataset** The ZINC dataset (Irwin & Shoichet, 2004) contains 12,000 small 2D molecular graphs with an average of 23 nodes with information on the node attributes and the edge attributes. The task is to regress the constrained solubility $\log P$ - SA - #cycle where $\log P$ is the octanol-water partition coefficients, SA is the synthetic accessibility score, and #cycle is the number of long cycles. For each molecular graph, the node features are the species of heavy elements.

**Model** The model is made up of six layers. Only the first two layers are MFN layers (in order to save on the number of parameters). The initial edge features $e_{ij}^{(0)}$ are obtained by concatenating PDF Yang et al. (2023) descriptors used for ZINC and the input edge features of the data set. At each layer $t$, the initial nodes features, $h_i^{(t)}$, are computed using a convolutional GNN with 140 channels. For the first two layers, the matrix is then formed using a multilayer perceptron (MLP) of size $[560, 16]$ with the GELU activation function and a batch norm,

$$H_{ij,c} = \text{MLP}(h_i^{(t)}, h_j^{(t)}, e_{ij}^{(t)}) \tag{32}$$

We compute the matrix function, using 16 poles and 16 channels. We extract the diagonal elements to update the node features, and use the off diagonal elements to update the edge features by passing them into a residual 2 layers MLP of size $[512, 140, 140]$. We used an average pooling followed by a linear layer for the readout.

**Training** Models were trained with AdamW, with default parameters of $\beta_1 = 0.9$, $\beta_2 = 0.999$, and $\epsilon = 10^{-8}$ and a weight decay of $5e - 5$. We used a learning rate of 0.001 and a batch size of 64. The learning rate was reduced using an on-plateau scheduler.

**Baselines** The baseline models used for the ZINC dataset comparisons include: GCN (Kipf & Welling, 2017b), GAT (Veličković et al., 2018), MPNN (Gilmer et al., 2017), GT (Dwivedi et al., 2020), SAN (Kreuzer et al., 2021), Graphormer (Ying et al., 2021), PDF (Yang et al., 2023)

## A.9.3 TUDATASETS

**Dataset** We train on a subset of the TUDatasets including the MUTAG, ENZYMES, PTC-MR, PROTEINS and IMDB-B subsets. The MUTAG, ENZYMES, PTC-MR, and PROTEINS datasets are molecular datasets that contain node-level information on the molecules. The IMDB-B a social network datasets.

**Model** The number of layers for each subset is in Table 6. We use MFN layers only for the first 3 layers (in order to save on the number of parameters). The number of hidden channels, $n_{\text{features}}$, for each model is given in 6. The initial edge features $e_{ij}^{(0)}$ are obtained by concatenating PDF Yang et al. (2023) descriptors used for each subset and the input edge features of the data set. In each layer $t$, the initial node features, $h_i^{(t)}$, are computed using a convolutional GNN. For the first two layers, the matrix is then formed using a multilayer perceptron (MLP) of size $[4 \times n_{\text{features}}, 16]$ with the activation function GELU and a batch norm,

$$H_{cij} = \text{MLP}(h_i^{(t)}, h_j^{(t)}, e_{ij}^{(t)}) \tag{33}$$

We compute the matrix function, using 16 poles and 16 channels. We extract the diagonal elements to update the node features, and use the off diagonal elements to update the edge features by passing them into a residual

2 layers MLP of size $[512, n_{\text{features}}, n_{\text{features}}]$. We used an average pooling followed by a linear layer for the readout.

**Training** Models were trained with AdamW, with default parameters of $\beta_1 = 0.9$, $\beta_2 = 0.999$, and $\epsilon = 10^{-8}$ and a weight decay of $5e - 5$. We used a learning rate of 0.001 and a batch size of 64. The learning rate was reduced using an on-plateau scheduler.

Table 6: Model training parameters. For the matrix functions the number of poles ($n_p$) and matrix channels ($c$) are indicated

| dataset | epochs | lr | $n_{\text{layers}}$ | $n_{\text{features}}$ | other |
|---------|--------|-----|---------|----------|-------|
| **MUTAG** | 301 | 1e-3 | 6 | 256 | $n_p$=16,$c$=16 |
| **ENZYMES** | 201 | 1e-3 | 6 | 256 | $n_p$=16,$c$=16 |
| **PTC-MR** | 151 | 1e-3 | 6 | 128 | $n_p$=16,$c$=16 |
| **PROTEINS** | 451 | 1e-3 | 6 | 128 | $n_p$=16,$c$=16 |
| **IMDB-B** | 301 | 1e-3 | 3 | 256 | $n_p$=16,$c$=16 |

**Baseline** The baseline models used for the TU datasets comparisons include: GK (Shervashidze et al., 2009), RW (Vishwanathan et al., 2010), PK (Neumann et al., 2016), AWE (Ivanov & Burnaev, 2018), PSCN (Niepert et al., 2016), ECC (Simonovsky & Komodakis, 2017), DGK (Yanardag & Vishwanathan, 2015), Caps-GNN (Xinyi & Chen, 2019), GIN (Xu et al., 2019), $k$-GNN (Morris et al., 2019), IGN (Maron et al., 2018), PPGNN (Maron et al., 2019), GCN$^2$ (de Haan et al., 2020) GraphSage (Hamilton et al., 2017), PDF (Yang et al., 2023)

