# OpenReview forum: "Equivariant Matrix Function Neural Networks"
_ICLR.cc/2024/Conference — ICLR 2024 spotlight_

### Official Review · Reviewer_Qi1r · 2023-10-19

**Soundness:** 2 fair
**Presentation:** 1 poor
**Contribution:** 3 good
**Rating:** 6
**Confidence:** 3

**Summary:**

GNNs need to balance the contributions from local (nearby nodes) data and more non-local data. The authors here present an approach based on spectral graph convolutional methods which can capture local and non-local features through a parameterization via resolvent expansions. The authors propose various means of implementing their method to both improve performance and speed up the technique. Their results are backed by experiments on a few datasets which show relatively good performance of the method.

**Strengths:**

The most important strength is that the paper presents a method which performs well on numerical experiments. Though I have some concerns about the experiments, if the results are confirmed and these concerns are appropriately addressed, I think this method should be documented and tested further.

Separate from the experiments, the proposed method is an extension of spectral convolutional approaches which have enjoyed great success. The particular strategy that the authors propose, via resolvent expansions, seems like a decent approach to take when including long range interactions. It is tied together relatively nicely with geometric data as well. Implementation can be tricky in certain parts (e.g. implementing complex numbers) but the architecture and overall idea is relatively sound.

**Weaknesses:**

The method is laid out in section 4 and to be honest, I had quite a bit of trouble following along. A number of notational concerns are pointed out below. But let me just start at the beginning and make sure I have the right picture. As I understand, the authors are choosing $b$ basis operators which are functions taking inputs consisting of a set of $\mathcal{X} \times V$ pairs where $V$ is some space for the hidden state. Then, they apply linear maps on top of these outputs from the bases which respect the graph permutation equivariance, hence a $bn \times bn$ matrix. I still am not sure this is the right interpretation, but the authors need to state this more formally and cleanly. They should specify what the functions $\phi_m$ do. I.e. specify input and output space. Indexing of eq. (7) is confusing. I believe $ij$ refers to the node indexing, but the matrix is $bn \times bn$ so $H_{cij, m_1, m_2}$ is not consistent with how the entries of the matrix work. Also, eq. (7) seems to indicate the entries are outputs of fixed basis functions, but these can be learnable functions as well? The authors should also give examples of $\phi_m$ as promised (is (21) an example of such?). I also believe that eq. (7) should be indexed over $k$ in the intersection of the neighbors, not the union. This is just a sampling of the confusions that arise and eventually required me to make decisions as to what is actually happening.




Moving on, I have concerns about the experiments. First, there is no code shared so I cannot verify numbers. What is shared in the zip file appears to be a dataset and some details on how to train a model for that dataset. Sharing code is obviously not a requirement, but it means I have to judge the result solely on what is written. Here, the methods of comparison to other results are potentially in question. For example, it appears complex numbers are being counted as a single parameter per number whereas it should be two per number (real and complex parts are both variables); see question below asking to clarify this. Runtime comparisons are also not provided, and this can be a concern here, since the method requires spectral decompositions (or approximations thereof). Finally, see questions below, but I am not sure how certain features are being extracted. E.g., the authors say that they take descriptors from the PDF model in the appendix for the ZINC data, but this would be unfair if they take features from a trained model on the same dataset. Nonetheless, if these issues are resolved, results look good as stated and if confirmed, this would add support to the method.

I simply could not follow section 4.3. What is the Combe-Thomas theorem? There is no citation for this. Also, doesn’t going from the bound stated in the paragraph before Eq (16) to the bound in Eq. (16) require some dependence on the norm of the weights? Later on, the geometric expressiveness statements are stated without proof and missing too many details for me to follow. I have many questions. First, what exactly are the authors saying is equivalent to the infinite layer MPNN (does the matrix channel need an infinitely sized pole expansion)? Where are the formal proofs of these statements? Does expressiveness here imply approximating a given function and if so, in what norm? What are two-body entries?

Notation comments:
- Eq (3): $\mathcal{G}$ in the notation was defined as a specific graph; however, here it is used to denote the space of graphs.
- Eq (4): $\Phi \odot g$ is confusing. Why is there a representation for the output space but not the input space?
- Below Eq (8): $\rho$ is bold in the equation but not bold there. Also, $\rho$ is unitary in general, not orthogonal as far as I understand. If it is orthogonal, then why is there a complex conjugate transpose instead of just a transpose?
- ... and many more. I would ask the authors to go through the paper and significantly revise and clean things up.


Altogether, I cannot recommend acceptance for this paper given the writing quality and concerns laid out above. Nonetheless, I welcome the authors' responses to these points.

**Questions:**

Other questions/concerns:
- When reporting parameter counts, the authors have some parameters which are complex numbers. Are these complex numbers counted as two parameters per number or one? A fair comparison should count each complex number as two parameters since there is a trainable real and complex part.
- It seems this parameterization is a nonlocal form of message passing and the resulting layers manipulate the eigenvalues of the Laplacian or adjacency matrix. We know from expressiveness results (WL or otherwise) that this is limited because the eigenvalues are insufficient to encode all the information in a graph. Is my understanding here correct? If so, what are the practical limitations that arise from this implementation?
- What are the $\phi_m$ basis functions for pure graph inputs?
- If the $\phi_m$ are being learned, did you include this in the parameter count?
- When you say "The initial edge features ... are obtained by concatenating PDF [Yang et al. (2023)] descriptors used for ZINC”, are these descriptors the outputs of the trained model? That would be unfair for comparison if so.

---

> ### Author Response · Authors · 2023-11-17
>
> Thank you very much for your review. We appreciate that you found that our method "performs well on numerical experiments" and works "nicely with geometric data". In our revision, we made a significant effort to make the method more understandable. In the following, we respond to your questions and suggestions to further improve the paper. We respectfully hope that our responses will be satisfactory and increase your score.
>
> ### Relationship to spectral convolutions
>
> > The proposed method is an extension of spectral convolutional approaches which have enjoyed great success. The particular strategy proposed by the authors, via resolvent expansions, seems like a decent approach.
>
> We want to emphasize here that there are key differences between Matrix Function Networks and previous spectral convolution approaches.
> - In all spectral convolution approaches known to us, the convolution operators were used as filters. We believe that the step of parameterizing features in a neural network architecture to be learnable graph operators and updating the node features based on matrix functions of these operators is new. To be precise, previous spectral convolution approaches have an update rule of \$X^{t+1} = f(L)X^{t}\$ where \$L\$ is fixed (the Laplacian), while MFN acts as \$X^{t+1} = f(H(X^{t}))\$ where both \$H\$ and \$f\$ are learnable. In particular, each entry in the learnable matrices is a many-body function of the local environment of each edge. In subsequent layers, new operators will be learned based on these extract features.
> - Furthermore, no spectral convolution methods known to us included equivariance to Lie groups, and therefore these methods were not competitive for geometric point clouds. Our method to parametrize the operators naturally expend to geometric point clouds.
>
> ### Clarification on the matrix construction step
>
> > The method is laid out in section 4 and to be honest, I had quite a bit of trouble following along.
>
> We understand that we attempted to make Section 4 the most general possible. However, we now realize that this leads to a very abstract and abstruse definition. We have rewritten the matrix construction step to simplify the language, and we added a new section that gives examples of concrete implementation of the matrix construction in the case of the \$O(3)\$-group. We also have largely expanded Figure 1 to provide a better illustration of the method, including a visual representation of each step of the matrix function network (see https://ibb.co/2NhsmRb for a preview). We hope that these edits will enable a better understanding of the key contributions of our method. Here is a more condensed version of the key part of our approach, simplifying a bit the notation:
> - First, the method takes a graph with node labels and edge labels as entries and constructs **local** many-body node features using a graph neural network layer (for example, a convolution graph neural network for pure graphs, or an equivariant graph neural network for geometric graphs), that we call here \$V_i^{(t)}\$, \$V_j^{(t)}\$.
> - Second, a graph matrix \$H_{ij}\$ is learned by parameterizing its entries with a learnable function of the node features \$H_{ij} = \phi(V_i^{(t)}, V_j^{(t)})\$. The notion of the size of the basis is relevant only for geometric graphs, where a Lie group is acting, because this function \$\phi\$ needs to be equivariant.
> - The node features are updated using the output of a learnable matrix function of these matrices. We want to stress that this is a **key** novelty in the context of GNNs. The output features are now **non-local** features.
>
> ### On the experiments
>
> > First, there is no code shared [...] For example, it appears complex numbers are being counted as a single parameter per number whereas it should be two per number, see question below asking to clarify this. [...] I am not sure how certain features are being extracted. For example, the authors say that they take descriptors from the PDF model in the appendix for the ZINC data.
>
> We have attached a copy of the two codes used to run the experiments, both for the Cumulene case and the pure graph case. The parameter count is correct; see the response below for more details, but only the poles are complex parameters. The PDF descriptors are **not** extracted from a trained model; we just use the Laplacian and fixed powers of the Laplacian of the graph to extract initial edge labels to enrich the information in pure graphs as initial embedding. We want to stress that this does not affect the number of parameters, and this extraction was not trained in any way. They are not related to **anything** learnable, just a fixed precomputable initial embedding. As our MFN learns custom operators beyond the Laplacian, we wanted to include this operator as a baseline. Sorry for the confusion this has generated.

---

> ### Author Response · Authors · 2023-11-17
>
> ### On the Combes Thomas Theorems
>
> > I simply could not follow section 4.3. What is the Combe-Thomas theorem?  Also, doesn’t going from the bound stated in the paragraph before Eq (16) to the bound in Eq. (16) require some dependence on the norm of the weights?
>
> We understand that this section was very technical and we moved most of this to the appendix and used the available space to clarify Section 4.1 on the core of the architecture. We now cite the Combes-Thomas theorem paper [1].
>
> Thank you for highlighting the dependence on the norm of the weights, this is an important point that we forgot to mention. We now added a brief explanation that the bounds can be taken uniform for suitable bounds on the operator which are guaranteed by our normalization procedure.
>
>
> ### Runtime comparison
>
> > Runtime comparisons are also not provided, and this can be a concern here, since the method requires spectral decompositions (or approximations thereof).
>
> We agree that computational cost is a key parameter in any method. As explained in Section 4.2 in the Linear Scaling Paragraph, the prototype implementation has a cubic scaling and is therefore quite slow compared to other approaches (a factor of x2 to x5 compared to local models). However, fast inversion methods are well established, and we explain how this scaling can be reduced significantly to less than quadratic scaling, all the way to linear scaling, by exploiting the sparsity of the matrices we construct and selected inversion methods.
>
> Although these methods are known, they require significant effort to implement and integrate them into machine learning libraries like Pytorch due to the need for specialized CUDA kernels. We intend to make this effort and release open-source code in future work, but we believe that this is beyond the scope of this paper, where we focus on the novelty of the architecture and its expressivity.
>
> Finally, we stress that one of the key results of this paper is that MFN can model systems that no other architectures can model, including Transformers. Due to the unstructured non-locality of global attention, Transformers cannot extrapolate to large systems in the Cumulene example. This means that regardless of speed, there are applications in which MFNs are clearly the only solution to date. We want to stress that this solves an important open problem in the community of machine learning force fields (see [3]). We made changes to the contributions to better highlight this key point.
>
> ### On relationship with infinite layer MPNN
>
> > Later on, the geometric expressiveness statements are stated without proof and missing too many details for me to follow. First, what exactly are the authors saying is equivalent to the infinite layer MPNN (does the matrix channel need an infinitely sized pole expansion)? Where are the formal proofs of these statements? Does expressiveness here imply approximating a given function and if so, in what norm? What are two-body entries?
>
> Although we agree that a full proof of this statement would be interesting, we believe that it is beyond the scope of this paper due to the very limited length. That is why we did not claim any proof but intended to present that as a remark for future investigation. We moved this part of the paper to the appendix and used the gain space to clarify other sections. We also have toned down the statement to show that it is a well-motivated remark rather than a theorem. Here, we give an intuition of the relationship between the two approaches:
> Any analytic matrix function \$f\$ admits a formal power series expansion, valid in its radius of analyticity,
> \begin{equation}
>     f(H) = \sum_{k=0}^{\infty} c_{k} H^{k}
> \end{equation}
> where \$c_{k}\$ are the complex (or real) coefficients of the expansion. Lets look at the diagonal elements of each of these powers,
>
> $(H)^{2}_ {ii, c00} = \sum_ {j \in \mathcal{N}(i)} H_ {ij,c0m}  H_ {ji,cm0}, \quad (H)^{3}_ {ii, c00} = \sum_ {j \in \mathcal{N}(i)}  \sum_ {k \in \mathcal{N}(j)} H_ {ij,c0m} H_ {jk,cmm'} H_ {ji,cm'0}$
>
> The sum over \mathcal{N}(i) is forced due to the sparsity of the constructed matrix \$H\$ in our method. Therefore, we observe a convolutional structure similar to message-passing methods. The power \$2\$, corresponds to a single convolution, the power \$3\$ to two convolutions, all the way to infinity. Each coefficient of the matrix function that is **learnable** can act like weights in the linear update of the analogous layer. Because the power series expansion is infinite, one sees a clear analogy between the two.
>
> ### On the parameters count
>
> > Are these complex numbers counted as two parameters per number or one?
>
> Thank you for your inquiry; all parameters were appropriately counted. We want to stress that only the poles of the resolvent expansion are complex parameters, the rest being real. The poles usually represent at most 256 parameters (times two for real and complex), which is negligible in parameter counts.

---

> ### Author Response · Authors · 2023-11-17
>
> ### On the expressiveness and WL test
>
> > It seems this parameterization is a nonlocal form of message passing and the resulting layers manipulate the eigenvalues of the Laplacian or adjacency matrix. We know from expressiveness results (WL or otherwise) that this is limited because the eigenvalues are insufficient to encode all the information in a graph. Is my understanding here correct?
>
> Thank you for this interesting question. First, we emphasize that MFN layers do not manipulate the eigenvalues of the Laplacian or adjacency matrix. We learn operators that are arbitrary operators. In particular, when we apply the MFN for geometric graphs, the learned operators are group equivariant, while the Laplacian or adjacency would just be invariant.
> For expressiveness, there are two separate interesting cases:
>
> - The first case is about expressiveness on pure graphs (not embedded in a vector space). In this case, it is true that many graphs share the same spectrum for the Laplacian or for the adjacency. However, the MFN learns **multiple** graph operators. Therefore, a well-converged MFN layer can learn a set of operators whose spectrum maximally discriminates graphs relevant to the learning task. Moreover, the fact that multiple operators are learned, even if some graphs are cospectral for one of the operators, means that they need not be cospectral for the union of the spectrums of all the operators. Therefore, an MFN layer should be more expressive than simply using the Laplacian spectrum.
> - The case for geometric graphs is different. The question is related to the geometric WL-test [1]. We know that if the matrix entries are learnt using a $G$-MACE [2] layer with correlation order equal to the number of points of the graph, then the spectrum of this operator is complete, in the sense that it will be able to discriminate any graph that are not related by permutations or symmetries of $G$.
>
> ### Basis functions
>
> > What are the basis functions for pure graph inputs? If they are being learned, did you include this in the parameter count?
>
> We have added a more detailed explanation of the matrix construction in pure graph that was previously in the appendix. Here is a short summary, using the new notation introduced in Section 4 to clarify our architecture. We use a convolutional graph neural network to learn the node features $V_{i}^{(t)}$ for each node $i$. We then use a simple multilayer perceptron $H_{ij,c} = \text{MLP}(V_{i}^{(t)}, V_{j}^{(t)}, e_{ij}^{(t)})$, where $e_{ij}^{(t)}$ are a concatenation of fixed edge attributes (see the PDF explanation) and learnable edge features extracted from the off-diagonal blocks of the matrix function. This was counted in the parameter count. We hope that the code will help clarify these points.
>
> [1] On the Expressive Power of Geometric Graph Neural Networks, Joshi, et al
> [2] A General Framework for Equivariant Neural Networks on Reductive Lie Groups, Batatia, et al
> [3] Machine learning force fields, Oliver T Unke, et al

---

> > ### Comment · Reviewer_Qi1r · 2023-11-21
> >
> > Dear authors,
> >
> >
> > Thank you for the detailed responses to my questions and feedback. It seems that many of my concerns are directly addressed:
> > - Parameter counts being undercounted due to having complex numbers is largely negligible. This appears to add at most about a thousand parameters. Nonetheless, I would still ask that the authors report the correct parameter count in the paper rather than saying certain parameters are negligible in count.
> > - The inputs are not trained outputs of PDF.
> > - Some code has been shared so I thank the authors for finally sharing it. I tried running it and personally had issues. This is likely due to my lack of time in going through the code, and wish I had more time to go through it.
> > - Many of the confusing notation and incorrect equations have been cleaned up. Nonetheless, I still have struggles to follow at times. For example, before equation $7$ we are told that the matrix $H$ is $MN \times MN$ but then the indexing $H_{ij,cm_1m_2}$ which adds a channel dimension and also makes the indexing in some sort of tensor notation where $ij$ comes before $m_1, m_2$ (i.e. this is clearly not the actual entry locations). This again leaves me having to make decisions about how this method works.
> >
> > Some things that are still concerns and I feel can be addressed better:
> > - Runtime comparisons are important here especially given the expensive matrix operations. The authors are clear that this slows things down, but still, a comparison should be done to see the trade-offs. After all, other networks also use expensive steps so the added time may or may not be as bad as stated. We can only tell once it's reported.
> > - Many parts of the discussion are still rather ad hoc and imprecise mathematically. For example, the *linear scaling cost* subsection makes a number of claims that may or may not hold in practice or are not cited well enough to make it clear that they are true. E.g. topological dimension often does not correlate with the actual sparseness and even if improvements in scaling can be made, this often comes with sources of error that are not accounted for. Similar concerns arise elsewhere too with the technical statements.
> > - As mentioned in my initial review, the experimental results seem promising. One drawback that is even more reinforced is the many complications that are added with this methodology: choice of feature constructors, integration of complex numbers, LDL factorizations, etc. I wonder how much of this is really needed. E.g. the authors could compare their resolvent method with simpler low order matrix functions (e.g. Chebyshev polynomials as they state) that do not need many of these complications.
> >
> >
> > On reflection, my perspective is that I still like the basic idea of the method. Though I am more convinced by the experiments, I think more detailed comparisons and additional analysis would help as mentioned earlier. The largest concern that still remains though is the writing quality. Many changes were made and though I could not review again the paper in full detail, I still was confused by parts and believe the technical quality is still missing at points. If this were a journal setting, I would ask the authors to revise and resubmit, but since that is not an option, I will improve my score to a weak accept and leave the ultimate decision up to the AC.

---

> ### Author Response · Authors · 2023-11-22
>
> We thank the reviewer for iterating on the feedback and helping us improve the quality of the submission.
>
> Based on this feedback, we have further revised the manuscript to use improved notation for the matrix indices. Additionally, we have added a section in the appendix on runtime comparison, which includes timings for the MFN models and the MACE model on the cumulene example. We have also added the following reference in the 'Linear Scaling' paragraph [1], providing a more formal exposition of the statement presented in the text.
>
> [1] T. A. Davis, 'Direct Methods for Sparse Linear Systems'

---

### Official Review · Reviewer_2GgR · 2023-10-31

**Soundness:** 4 excellent
**Presentation:** 3 good
**Contribution:** 3 good
**Rating:** 8
**Confidence:** 4

**Summary:**

This paper introduces a novel architecture for processing graph data. The goal of the proposed method is to model non-local interactions between nodes in graphs without the shortcomings of alternate approaches. Namely, over-smoothing and excessive compute requirements for message-passing GNNs and the need for hand-designed features with spectral GNNs.

Matrix function neural networks learn equivariant representations over graphs. The approach consists of three stages: 1) building a matrix that corresponds to a group-equivariant matrix operator 2) applying a learned matrix function to this matrix 3) updating the matrix. To my understanding, the major contributions appear in the second part. The authors make use of resolvent calculus to determine an efficient parameterization of the matrix function. Given that the underlying matrix is sparse, this parameterization admits an efficient (potentially linear) algorithm for computing the function.

The method is evaluated on three different graph learning datasets, where it generally outperforms the baseline methods (or performs at least as well).

**Strengths:**

To my knowledge, this is an original and novel approach to designing GNNs. The authors describe theoretical and practical differences between their approach and existing methods. The proposed approach pulls on learnings from several other areas of study (resolvent calculus, electronic structure methods, and more) to design an expressive and efficient class of graph neural networks. The resulting method is elegant and performs well in the empirical evaluation.

Overall, I found the paper to be clear and well-written. I am not familiar with resolvent calculus or some of the other technical background introduced in section 4, but I was able to understand and follow the details. Furthermore, full details of the experiments are provided in the supplementary.

I feel that the contributions in this work are significant. GNNs are a busy area of research and it is no small feat to propose a novel and practical architecture. Empirically, the proposed method performs well. While the margin of improvement is often narrow, MFNs consistently achieve performance at least on par with leading methods in comparison.

**Weaknesses:**

It is unclear whether MFNs can be used as a drop-in replacement for GNNs. In two of the experiments, the models used are not end-to-end matrix function networks, due to model size considerations. Instead, a small number of MFN layers are used with more standard layers following. In all experiments, initial edge features are also extracted from other architectures --- I haven't quite understood the implication of this in terms of comparison to baselines.

The method is fairly complex with several design decisions. For example, the choice of basis, choice of update method, number of learnable weights and poles, etc. Moreover, making the method computationally tractable requires some techniques that are unfamiliar to the learning community at large. Therefore, I expect that the method may be adopted quite slowly by the community. This could be alleviated with an open-source implementation.

**Questions:**

One area of the paper that I feel I couldn't completely understand was on the introduction of a choice of basis (Section 4.1). It is stated that this choice should be made with consideration of the application domain. On the surface, this seems to align with spectral GNNs that lean on hand-designed features for effective learning. However, the authors point out that this can be approximated well in some cases. I'm curious to know what the limitations here are. When are we unable to automatically learn these functions? And are there any practical considerations for learning these?


Do we need to ensure that the learnable weights and poles satisfy any constraints for the resolvent calculus definition (Section 4.2)? From reading the explanation in the paper, I was under the impression that the choice of weights and poles determine the curve implicitly via the quadrature rule. Is it possible that we could learn weights and poles that lead to invalid curves, or otherwise undesirable behaviour?

As I mentioned above, the MFN models used in experiments always contain only a small number of layers. Was this a computational consideration?

Minor comments:

- There are a couple of symbols that are not defined in the main text. These can be mostly be inferred from context, but should ideally be defined.
    - Above equation (1), the domain and codomain of $s$ should be defined.
    - I could not find a definition of $\mathfrak{Z} z$ and related symbols used in Section 4.3.
- You cite transformers as having a problematic quadratic scaling. While this is true in the original formulation, it is quite common nowadays to use methods with far more efficient scaling. For example, PerceiverIO can reduce the computational cost to linear.
- Shaw et al. 2018 are cited for positional encodings, but these are also used in the original Vaswani et al. transformer paper.

---

> ### Author Response · Authors · 2023-11-17
>
> Thank you very much for your review. We appreciate that you found our work "novel" and "strong empirical evidence". In the following, we respond to your questions and suggestions to further improve the paper.
>
> ### On the basis
>
> > It is stated that this choice should be made with consideration of the application domain. I'm curious to know what the limitations here are. When are we unable to automatically learn these functions? And are there any practical considerations for learning these?
>
> Thank you for an excellent question. In the case of pure graphs, enumerating all possible graph operators is NP-hard. The approach we take to do the parameterization in a data-driven way takes the assumption that the model would converge to learn the optimal set of operators for the specific task. There are no limitations in the parameterization except that the matrix needs to be permutation invariant. However, for a graph with N nodes, the entry of the matrix might be an N-body function of all the other nodes. In our experiments, we simply used a multilayer perceptron to learn these entries, but it is likely that other solutions perform better. There are many theoretical open questions related to learning these operators, linked to the cospectrality to a set of spectrums of operators. In the case of geometric graphs (which are embedded in a vector space), the answer can be made more complete. In this case, the entries are \$G\$-equivariant functions of the graph. The proof of completeness of equivariant architectures like ACE or MACE for general Lie groups shows that for an infinitely large basis, any of these functions can be uniformly approximated with arbitrary accuracy. \[1,2\]
>
> ### Conditions on the poles
>
> > Is it possible that we could learn weights and poles that lead to invalid curves, or otherwise undesirable behaviour?
>
> Yes, poles that are too close to being purely real lead to divergences. Normalization of the matrix was fundamental to ensure that the optimizer did not explore these points. In practice, finding a good parameterization of the poles was important to making the method work. We parametrize the poles as \$z_{s} = w_{1}\exp(w_{2}) + w_{3}\$, with \$w_{1}, w_{2}, w_{3}\$ learnable weights initialized from a standard Gaussian. For the weights, conditions on the norm of the weights would mean targeting a specific space of functions. We are just initializing these weights with a Gaussian centered on zero and a variance one, and let the optimizer move them freely. However, we believe that more clever initialization based on some inductive bias might improve the convergence in applications where one knows the kind of matrix function that gives rise to non-locality (e.g. Fermi Dirac in electronic systems).
>
> ### Number of layers
>
> > As I mentioned above, the MFN models used in experiments always contain only a small number of layers. Was this a computational consideration?
>
> For the pure graph case, it was mostly for restricting the number of parameters and for computation consideration. We intend to implement specialized CUDA kernels for fast inversion as outlined is Section 4.2, that would enable us to scale to larger number of matrices. For the molecular case, we do not see any improvement when increasing the number of layers past 2. This is expected from previous work with many-body methods \[3\].
>
> ### Minor comments
>
> Thank you for pointing out these small notation issues. We changed to the more conventional $\text{Imag}$ to denote the imaginary part in Section 4.3. We also have added a reference to PerceiverIO, mentioning that better scaling can be achieved with newer implementations.
>
> \[1\] Atomic Cluster Expansion: Completeness, Efficiency and Stability, Dusson, et al
>
> \[2\] A General Framework for Equivariant Neural Networks on Reductive Lie Groups, Batatia, et al
>
> \[3\] MACE: Higher Order Equivariant Message Passing Neural Networks for Fast and Accurate Force Fields, Batatia, et al

---

> > ### Comment · Reviewer_2GgR · 2023-11-20
> >
> > I appreciate the author's thorough rebuttal. I have read through the other reviews, and your responses there too.
> >
> > I feel that all of my questions have been adequately answered. I am pleased that several improvements to clarity have been made to the paper.
> >
> > I am keeping my score, as I believe this to be a meaningful piece of work that provides a novel approach to graph learning and gives sufficient empirical evidence of the effectiveness of the proposed method.

---

### Official Review · Reviewer_qFqk · 2023-11-01

**Soundness:** 3 good
**Presentation:** 1 poor
**Contribution:** 3 good
**Rating:** 5
**Confidence:** 2

**Summary:**

This paper proposes MFN, a graph neural network where the intermediate representation takes a matrix form. The matrix form makes the neural network equivariant for a group action where the representation is unitary. An efficient computation based on the power expansion of the matrix is also proposed. The experiments on the molecule datasets show the competitive performance of MFN.

**Strengths:**

Strong empirical results.

Matrix variate formulation and its efficient approximation by the resolvent expansion are novel.

**Weaknesses:**

There is room for improvement in the presentation. I found several incomplete descriptions. For example,
- Given a matrix $A$, $A^*$ is not defined (I guess it's adjoint of A)
- $f_\theta(H)$ is defined as "scalar to scalar," but it seems "matrix to matrix".
- In Eq. (10), since $H$ is a matrix, it's not obvious how $g$ acts on $H$. I guess H is the function of $\sigma$ and $g$ acts on it, but it's not clearly described.
- I couldn't find the proof of Eq. (10).
Also, see the "Questions" area below.

It is not easy to grasp the computational cost of MFN since it depends on the graph dimension d, the choice of the update equations (11)--(13), etc. A clear comparison is needed. For example, you can evaluate the actual wall clock time and memory usage, which would be clear evidence that MFN is better than other methods, including Transformer.

**Questions:**

In experiments:
- What is L in the caption of Fig 2?
- It seems H is constructed with Kronecker products as in Eq. (21). What is the actual size of H?

---

> ### Author Response · Authors · 2023-11-17
>
> Thank you very much for your review. We appreciate that you found our work "novel" and with "strong empirical evidence". In the following, we respond to your questions and suggestions to further improve the paper. We respectfully hope that our responses will be satisfactory and will increase your mark.
>
> ### General remark
> > There is room for improvement in the presentation.
>
> We tried to make the architecture as general as possible. However, we now see that this abstraction made the method quite hard to grasp. We have now added several sections, including examples of the MFN architecture for specific groups, to exemplify the architecture in a concrete setting. We also have rewritten part of Section 4 to clarify the matrix construction step. Finally, we have improved Figure 1 to give a better illustration of the method. We hope that has significantly improved the overall presentation of the method.
>
> ### Some details
>
> > Given a matrix \$A\$, \$A^{*}\$ is not defined (I guess it's adjoint of A).
> > \$f_{\theta}(H)\$ is defined as "scalar-to-scalar," but it seems "matrix-to-matrix".
>
> \$A^{*}\$ is indeed the adjoint of \$A\$ in the paper. Thank you for spotting these; we have added clearer introductions to all the notation in the paper, including the adjoint. The ambiguous notation of the function \$f\$ is to highlight that any real value function can be "overloaded" to a symmetric matrix value function (matrix to matrix) by simply applying it point-wise to its eigenvalues. An alternative view if \$f\$ is analytic is to consider the formal power series expansion of the function \$f\$, and formally replace the scalar powers to matrix powers. In this context, it is widely accepted to use the same symbol for the real value map and the matrix value map (see, for example, the Wikipedia page on matrix functions [https://en.wikipedia.org/wiki/Analytic_function_of_a_matrix]()).
>
> ### Action of \$G\$ on the matrix \$\textbf{H}\$
> > In Eq. (10), since \$\textbf{H}\$ is a matrix, it's not obvious how \$g\$ acts on. I guess H is the function of and acts on it, but it's not clearly described.
>
> Yes, \$g\$ is a group action that acts on the state of each of the nodes. At the matrix level, the action is given by Eq. 9, since the functions \$\phi\$ are assumed to be equivariant. The action of \$G\$ propagates from the actions in the states to the action in the matrix. A good example is for 3D graphs. In the case \$G=O(3)\$ the group of 3D rotations and the element \$g\$ are rotation matrices \$R\$. The state of each node would be a 3D vector of positions \$(x, y, z)\$. Rotations can be applied to the graph by rotating each position \$(x, y, z) \to R (x, y, z)\$. The function \$\phi\$ being equivariant will result in matrix entries that are also equivariant. Therefore, if a rotation is applied to the whole graph, transforming by a rotation \$R\$, then there exists an orthogonal matrix \$D(R)\$ such that the graph operator \$H_{1}\$ of \$G_{1}\$ is transformed into \$H_{2} = D(R) H_{1} D(R)^{T}\$.
>
> ### Proof of equivariance of matrix function \$H\$
> > I couldn't find the proof of Eq. (10).
>
> We have written a detailed proof of the equivariance of generic continuous matrix functions in the appendix and reference it in the main text. Thank you for noticing that.
>
> ### Computation cost of MFN
> > It is not easy to grasp the computational cost of MFN since it depends on the graph dimension [...] For example, you can evaluate the actual wall clock time and memory usage, which would be clear evidence that MFN is better than other methods, including Transformer.
>
> We agree that computational cost is a key parameter in any method. As explained in Section 4.2 in the Linear Scaling Paragraph, the prototype implementation has a cubic scaling and is therefore quite slow compared to other approaches (a factor of x2 to x5 compared to local models). However, fast inversion methods are well established, and we explain how this scaling can be reduced significantly to less than quadratic scaling, all the way to linear scaling, by exploiting the sparsity of the matrices we construct and selected inversion methods.
>
> Although these methods are known, they require significant effort to implement and integrate them into machine learning libraries like Pytorch due to the need for specialized CUDA kernels. We intend to make this effort and release open-source code in future work, but we believe that this is beyond the scope of this paper, where we focus on the novelty of the architecture and its expressivity.
>
> Finally, we stress that one of the key results of this paper is that MFN can model systems that Transformers cannot model. Due to the unstructed non-locality of global attention, Transformers can not extrapolate to large systems in the Cumulene example. This means that regardless of speed, there are applications in which MFNs are clearly the only solution to date. We made changes to the contributions to better highlight this key point.

---

> ### Author Response · Authors · 2023-11-17
>
> ### Construction of \$H\$ in the \$O(3)\$ group
> > What is L in the caption of Fig 2?
> > It seems H is constructed with Kronecker products as in Eq. (21). What is the actual size of H?
>
> The \$L\$ index corresponds to the maximal spherical order in the basis expansion of the matrix for the spherical case. In order to make this notation clearer and improve the presentation of the paper, we have added a section to detail the implementation of the matrix construction case in the \$O(3)\$ group. We hope that this will significantly help the readers understand the method. \$H\$ is a tensor of size \[\$N_{\text{nodes}} \times (L+1)^{2}, N_{\text{nodes}}\times(L+1)^{2}, N_{\text{channels}}\$], where \$N_{\text{nodes}}\$ is the number of nodes in the graph, and \$N_{\text{channels}}\$ is a hyper-parameter corresponding to the number of operators we are learning.

---

> ### Author Response · Authors · 2023-11-22
>
> Dear Reviewer qFqk,
>
> Thank you again for your valuable feedback and comments!
>
> As the discussion period is ending soon, we would greatly appreciate it if you could let us know whether you are satisfied with our response. We will be happy to address any remaining concerns.
>
> Sincerely, Authors

---

### Author Response · Authors · 2023-11-17
**General reply**

We thank all reviewers for their time and effort in reviewing our paper.
We are encouraged by your recognition of our work's "strong empirical results" (R1) and that our proposed architecture is "sound" (R3) and "elegant" (R2), providing a "significant" (R2) contribution to the field of graph neural networks (GNNs).

We acknowledge that the presentation of our novel ideas had to be improved. To this end, we have made significant changes to improve the clarity of the manuscript, improving both figures and expanding on key ideas in the text. We hope that the key message is now much clearer.

Furthermore, in our original manuscript we did not sufficiently emphasize our architecture's ability to capture intricate non-local interactions in 3D point-clouds, as exemplified by the cumulenes in Figure 3 and Table 1. Our architecture can model systems that other, including transformer-based architectures, cannot even qualitatively reproduce (see Table 1). This result is a critical advancement within the field of machine learning forces and is, in our view, the most important result of this work, solving an important open problem in the community (see [1]). To this end, we improved our list of contributions and overall referencing of the results to emphasize this result.

Concrete enhancements to our manuscript include:

- Significant restructuring of Section 4.1, which introduces the architecture. In particular, we have largely expanded the section on the matrix construction step.
- Addition of a section with a concrete example of the construction of the matrix for the rotation group in 3D. The rotation group is where our main result (expressivity, as illustrated by the cumulene example) comes from. We hope that an explicit example makes the entire method clearer. We also added a figure to the main text to show the shape of the matrix in this specific case.
- The contributions paragraph and results presentation have been revised to emphasize the key finding of the paper: the ability of MFNs to model quantum systems where all previously published architectures fail to give even qualitatively accurate predictions.
- Expanding on Figure 1. to provide a better illustration of the method, including a visual representation of each step of the matrix function network (see https://ibb.co/2NhsmRb for a preview).
- Improvements in notation and general clarity of the writing.
- Expanding on the proof of the equivariance of the matrix function.
- Improving the section on computational cost and perspective for linear scaling with the method.
- The code is now available to reviewers (see `readme-mfn.md` for further information).

[1] Oliver T Unke, et al. Machine learning force fields. Chemical Reviews, 121(16):10142–10186, 2021.

Below, we respond to the reviewers’ comments individually.

---

### Author Response · Authors · 2023-11-17
**On the code**

We make the code available to the reviewers. It can be downloaded using the link: https://ufile.io/km8wech6 (press Free download -> Slow speed, and if a new window opens, please close it).

---

### Meta-Review · Area_Chair_ZtPC · 2023-12-12

**Metareview:**

This paper proposes a new framework for Graph Neural Networks using the concept of Matrix Functions. They are good in modelling non-local interactions, which can be the case especially with scientific and physical data. All reviewers appreciated the novelty and the empirical justification to a certain degree, although criticism regarding clarity and regarding completeness of experiments was raised by 1 out of 3 reviewers. The authors provided very detailed answers to the points raised, which led to higher scores after rebuttal, save for the one review that was not updated at all, and corresponded to a weaker confidence. The last reviewer appreciates the method and the paper and suggests more experiments, however. Overall, I suggest that this paper is accepted and I recommend the authors to closely follow the suggestions of reviewer Qi1r and implement them for the camera-ready.

**Justification For Why Not Higher Score:**

The paper could be further improved in terms of clarity and experimental completeness.

**Justification For Why Not Lower Score:**

Novel idea.

---

### Decision · Program_Chairs · 2024-01-16

Accept (spotlight)